# Structural insights into pathogenic mechanism of hypohidrotic ectodermal dysplasia caused by ectodysplasin A variants

Kang Yu [1,2,3,6], Chenhui Huang [2,3,6], Futang Wan[2,3,6], Cailing Jiang[1,2,3], Juan Chen[2,3], Xiuping Li[2,3], Feng Wang[4] ✉, Jian Wu [2,3] ✉, Ming Lei [2,3,5] ✉ & Yiqun Wu [1] ✉

EDA is a tumor necrosis factor (TNF) family member, which functions together with its cognate receptor EDAR during ectodermal organ development. Mutations of *EDA* have long been known to cause X-linked hypohidrotic dysplasia in humans characterized by primary defects in teeth, hair and sweat glands. However, the structural information of EDA interaction with EDAR is lacking and the pathogenic mechanism of *EDA* variants is poorly understood. Here, we report the crystal structure of EDA C-terminal TNF homology domain bound to the N-terminal cysteine-rich domains of EDAR. Together with biochemical, cellular and mouse genetic studies, we show that different *EDA* mutations lead to varying degrees of ectodermal developmental defects in mice, which is consistent with the clinical observations on human patients. Our work extends the understanding of the EDA signaling mechanism, and provides important insights into the molecular pathogenesis of disease-causing *EDA* variants.

Ectodermal dysplasia (ED) are genetic human disorders determined by developmental defects in tissues of ectodermal origin[1]. There are more than 200 different clinical types of ED, the most frequent subtype of which is hypohidrotic ectodermal dysplasia (HED) affecting approximately one in 5000–10,000 newborns[2]. HED is typically identified by three clinical characteristics, hypodontia (congenital absence of teeth), hypohidrosis (reduced ability to sweat), and hypotrichosis (sparseness of scalp and body hair), and may also be complicated with other secondary features such as dry fragile-appearing skin, periorbital hyperpigmentation and dry eyes[2,3]. HED can be inherited in an autosomal dominant, autosomal recessive, or X-linked manner, and the X-linked HED (XL-HED) that accounts for more than half of HED is caused by mutations in the *EDA* gene located at chromosome X[4–6].

The *EDA* gene codes for the ectodysplasin A (EDA) protein, a critical signaling factor involved in the interaction between the ectoderm and the mesoderm during embryonic development, regulating the establishment of placodes that bring about ectodermal organs including skin, hair, nails, teeth, and sweat glands[7]. EDA is a type II transmembrane protein with a collagen and a TNF homology domain (THD) in its extracellular portion that can be processed to a soluble ligand by cleavage at a furin protease site[8]. Several EDA isoforms are produced due to alternative splicing, and the two longest ones, EDA·A1 and EDA·A2, predominate and comprise about 80% of the total EDA proteins[9]. These two splice forms differ by a two-amino-acid motif, Val307-Glu308, that is only present in EDA·A1[10]. The crystal structures of both human EDA·A1 and EDA·A2 have been reported previously[11].

[1]Department of Second Dental Center, Ninth People's Hospital Affiliated with Shanghai Jiao Tong University, School of Medicine, Shanghai Key Laboratory of Stomatology, National Clinical Research Center of Stomatology, Shanghai, China. [2]Ninth People's Hospital, Shanghai Jiao Tong University School of Medicine, Shanghai 200011, China. [3]Shanghai Institute of Precision Medicine, Shanghai 200125, China. [4]Department of Oral Implantology, Ninth People's Hospital Affiliated with Shanghai Jiao Tong University, School of Medicine, Shanghai Key Laboratory of Stomatology, National Clinical Research Center of Stomatology, Shanghai, China. [5]State Key Laboratory of Oncogenes and Related Genes, Shanghai Jiao Tong University School of Medicine, Shanghai 200025, China. [6]These authors contributed equally: Kang Yu, Chenhui Huang, Futang Wan. ✉e-mail: diana_wangfeng@aliyun.com; wujian@shsmu.edu.cn; leim@shsmu.edu.cn; yiqunwu@hotmail.com

Similar to other TNF family members, the EDA trimers in both isoforms are formed by three jelly-roll β-sandwich monomers[11]. EDA·A1 and EDA·A2 interact with two distinct receptors EDAR and XEDAR respectively via binding of their THDs to the cysteine-rich domains (CRD$_S$) in their receptors[10]. It is unclear how the subtle difference between EDA·A1 and EDA·A2 leads to the binding to their respective receptors. Despite that both EDAR and XEDAR function via the activation of the NF-κB pathway, their extracellular regions, intracellular domains, and signaling pathways are all divergent, indicating that EDA·A1 and A2 might function in a different context[10]. In fact, it has been well established that EDA·A1 and EDAR are associated with HED, while EDA·A2 and XEDAR are unlikely involved[12]. EDA·A1 interacts with its receptor EDAR and functions together to regulate the initiation, morphogenesis, and differentiation during ectodermal organ development (Fig. 1a)[13].

To date, a large number of *EDA* mutations have been recorded in the Human Gene Mutation Database (HGMD, http://www.hgmd.cf.ac.uk/). Most of them are associated with HED, while some cause non-syndromic tooth agenesis (NSTA) that only affects the dentition[14]. Notably, substantial *EDA* missense mutations are located in the THD and thus might directly influence receptor binding or signaling capability[15]. However, the lack of structure information of the EDA-EDAR complex greatly hinders our mechanistic understanding about these disease-causing *EDA* variants.

In the present study, we report the crystal structure of the human EDA·A1$_{THD}$-EDAR$_{CRDS}$ complex, and reveal the important role of this complex in ectodermal development by using in vitro biochemical and cellular assays and in vivo mouse model analyses. Our findings help extend the understanding of the EDA·A1-mediated signaling

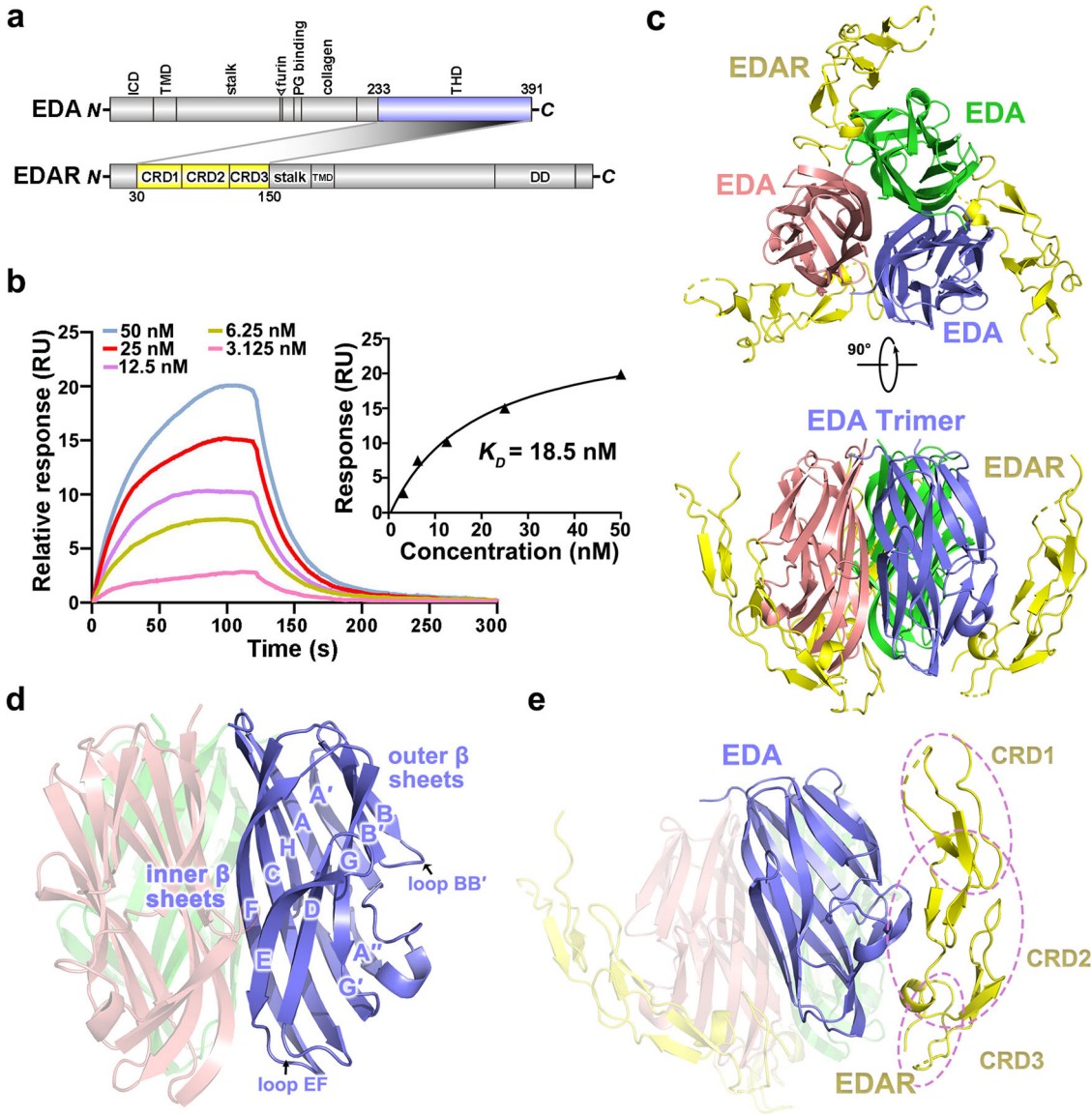

**Fig. 1 | Overview of the EDA·A1$_{THD}$-EDAR$_{CRDS}$ complex structure. a** Domain organization of EDA·A1 and EDAR. The interacting domains are labeled and highlighted in different colors. The shaded areas indicate the interaction between EDA·A1 and EDAR. ICD intracellular domain, TMD transmembrane domain, THD TNF homology domain, CRD cysteine-rich domain, DD death domain. **b** Surface plasmon resonance measurements showing that WT EDA·A1$_{THD}$ interacts with EDAR$_{CRDS}$ in a concentration-dependent manner. Graphs of equilibrium response unit versus EDA·A1$_{THD}$ concentrations are plotted. The estimated $K_D$ for the interaction is about 18.5 nM. **c** Top and side views of the EDA·A1$_{THD}$-EDAR$_{CRDS}$ complex. EDA·A1$_{THD}$ is a trimeric assembly (slate blue, salmon and green) and each EDAR$_{CRDS}$ (yellow) attaches to one side of the ligand. **d** Cartoon representation of the EDA·A1$_{THD}$ trimer. Loops and β strands of one EDA·A1$_{THD}$ monomer are labeled. **e** The structure of EDAR$_{CRDS}$ adopts an elongated conformation. CRD1, CRD2, and CRD3 are highlighted by dashed purple circles. Source data are provided as a Source Data file.

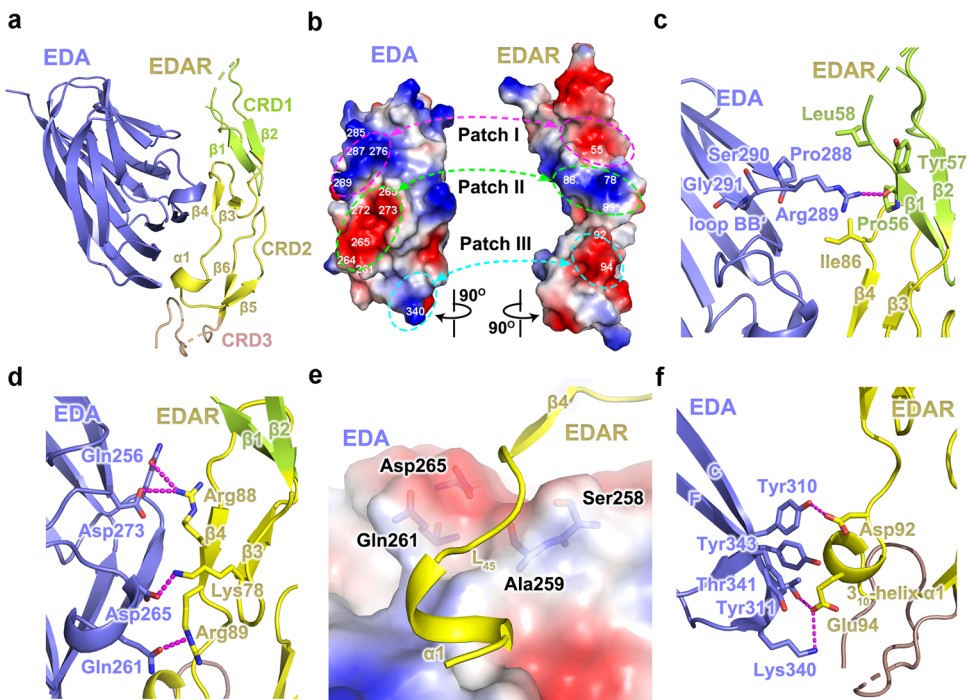

**Fig. 2 | Interactions between EDA·A1_THD and EDAR_CRDS. a** The interaction between EDA·A1_THD and EDAR_CRDS is mainly mediated by CRD2 of EDAR_CRDS. CRD1, CRD2 and CRD3 are highlighted in limon green, yellow and wheat, respectively. **b** The electrostatic surface potential of the binding site in EDA·A1_THD and EDAR_CRDS. Three pairs of surface patches with opposite electrostatic properties important for the ligand-receptor interaction are highlighted in dashed circles. Positive potential, blue; negative potential, red. **c, d** Detailed interactions at patches I and II between EDA·A1_THD and EDAR_CRDS. EDA·A1_THD and EDAR_CRDS are shown in cartoon representation. Residues involved in the interaction are shown in stick models and electrostatic interactions are denoted as magenta dashed lines. **e** Strand β4 and loop L_45 (between strand β4 and helix α1) in EDAR_CRD2 sit on the saddle-shaped surface of EDA·A1_THD. **f** Detailed interactions at patch III between EDA·A1_THD and EDAR_CRDS. EDA·A1_THD and EDAR_CRDS are shown in cartoons and interacting residues in stick models.

mechanism and provide structural insights into the pathogenesis of disease-related *EDA* variants in humans.

## Results

### Overall structure of the EDA·A1_THD-EDAR_CRDS complex

To understand how EDAR recognizes EDA, we first characterized the interaction between EDA·A1_THD (residues 233–391) and EDAR_CRDS (residues 30–150) using a surface plasmon resonance (SPR) assay, which revealed that EDA·A1_THD binds to EDAR_CRDS with an equilibrium dissociation constant ($K_D$) of 18.5 nM (Fig. 1b). This EDA·A1_THD-EDAR_CRDS interaction was further demonstrated by the comigration of EDA·A1_THD and EDAR_CRDS in a size exclusion chromatography analysis (Supplementary Fig. 1a).

To reveal the structural basis for the interaction between EDA·A1_THD and EDAR_CRDS, we crystallized the EDA·A1_THD-EDAR_CRDS complex and determined its structure using the single-wavelength anomalous dispersion (SAD) method at a resolution of 2.8 Å (Fig. 1c, Supplementary Fig. 1b and Supplementary Table 1). The crystal structure reveals that EDA·A1_THD and EDAR_CRDS form a heterohexameric complex, displaying a triangular-shaped architecture with an EDA·A1_THD trimer in the center and three EDAR_CRDS molecules at the vertexes of the complex (Fig. 1c). The EDA·A1_THD protomer has a β-sandwich structure containing two stacked β-sheets that adopt a classical jelly-roll topology with the inner sheets involved in the homotrimeric contacts while the outer β-sheets together with the bridging loops mediating the interaction with EDAR_CRDS (Fig. 1d). The structure of the THD trimer in the complex is almost identical to that of the crystal structure of the apo EDA·A1, indicating that binding to the receptor does not cause substantial structural changes of the ligand (Supplementary Fig. 1c)[11].

The extracellular region of EDAR consists of three CRDs and adopts an extended conformation that span a length of about 65 Å

(Fig. 1e). Each EDAR_CRDS molecular binds across the convex surface of a single EDA·A1_THD in the trimer, burying about 1340 Å² of total solvent accessible surface area at the intermolecular interface (Fig. 1e). The 1:1 binding mode between EDA·A1 and EDAR is markedly different from those previously observed in other TNF-receptor complex structures, such as TNF-TNFR2 and RANKL-RANK, in which each receptor binds into the concaved groove formed by two adjacent ligand protomers of the receptor (Supplementary Fig. 1d)[16,17].

### The EDA·A1-EDAR interaction

At the EDA·A1_THD-EDAR_CRDS interface, although all three CRDs of EDAR form direct contacts with EDA·A1_THD, the second CRD mediates most of the interactions with the ligand (Fig. 2a). The driving force for the binding of EDA·A1_THD to EDAR_CRDS is predominantly electrostatic interactions via three pairs of charged patches on the surfaces of the receptor and the ligand with complementary electrostatic potentials (Fig. 2b). The first patch of EDA·A1_THD interacts with the receptor through its BB' loop to strand β1 of EDAR_CRD1 (Fig. 2c). Upon binding to the receptor, the side chain of Arg289 in the BB' loop changes its orientation and extends into a negatively charged depression on CRD1, forming a salt bridge to the carboxyl group of Pro56^EDAR (Fig. 2c and Supplementary Fig. 2a). The second pair of patches contribute most of the electronic contacts between EDA·A1_THD and EDAR_CRDS. A cluster of acidic and polar residues in EDA·A1_THD (Gln256, Gln261, Asp265, Asp273) mediate a panel of electrostatic interactions with positively charged residues in EDAR_CRD2 (Lys78, Arg88, Arg89), tightly tethering antiparallel β3-β4 motif of EDAR_CRDS to the surface of EDA·A1_THD (Fig. 2d). Notably, at the boundary of this interface stand βA" of EDA·A1_THD forms a saddle-shaped depression, at which residue Ala259^EDA is the saddle point and Ser258^EDA and Gln261^EDA form the pommel and cantle of the saddle (Fig. 2e and Supplementary Fig. 2b). This unique structure snugly holds the β4 strand of EDAR_CRD2 by both

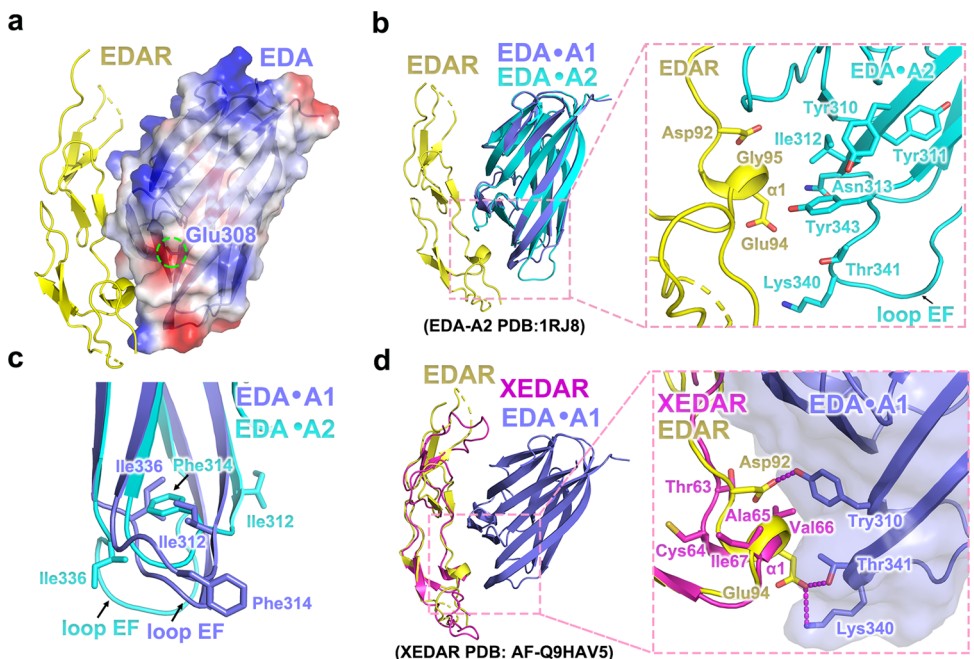

**Fig. 3 | The interaction specificity between EDA·A1$_{THD}$ and EDAR$_{CRDS}$. a** Residue Glu308 in EDA·A1$_{THD}$ exhibits no direct interaction with EDAR$_{CRDS}$. EDA·A1$_{THD}$ is shown in electrostatic potential surface representation with residue Glu308 denoted by a dashed green circle. The EDAR$_{CRDS}$ is shown in cartoon mode. **b** Structural comparison of EDA·A1$_{THD}$ and EDA·A2$_{THD}$. The structure of EDA·A2$_{THD}$ (PDB: 1RJ8) is superimposed onto the EDA·A1$_{THD}$-EDAR$_{CRDS}$ structure. A close-up view of the interface centered at helix α1 of EDAR$_{CRD2}$ is shown on the right.

EDA·A1$_{THD}$, EDA·A2$_{THD}$ and EDAR$_{CRDS}$ are colored in slate blue, cyan, and yellow, respectively. **c** Structural comparison showing the positional change of Phe314 and Ile336 in EDA·A2$_{THD}$. **d** Superposition of the predicted XEDAR$_{CRDS}$ generated by AlphaFold (PDB: AF-Q9HAV5) onto the EDA·A1$_{THD}$-EDAR$_{CRDS}$ complex. A close-up view of the interface centered at helix α1 of XEDAR$_{CRD2}$ is shown on the right. EDA·A1$_{THD}$, EDAR$_{CRDS}$, and XEDAR$_{CRDS}$ are colored slate blue, yellow, and hotpink, respectively.

shape and charge complementarity, defining the path of the loop C-terminal to strand β4 that guides the rest of EDAR to the third charged patch on the surface of EDA·A1$_{THD}$ (Fig. 2e and Supplementary Fig. 2b). The contact area on the third patch of EDA·A1$_{THD}$ is mainly contributed by a panel of residues with large aromatic or basic side chains, Tyr310 and Tyr 311 from βC and Lys340, Thr341 and Tyr343 from βF, which surrounds the acidic protrusion formed by the 3$_{10}$-helix α1 in EDAR$_{CRD2}$ (Fig. 2f). In particular, the side chains of Tyr311, Lys340, and Thr341 constitute a basic depression that nicely accommodates the side-chain carboxylic acid group of Glu94$^{EDAR}$ via direct electrostatic interactions with Lys340 and Thr341 of EDA·A1$_{THD}$ from opposite directions, impacting on the position and orientation of CRD2 helix α1 (Supplementary Fig. 2c).

Besides the aforementioned EDA·A1$_{THD}$-EDAR$_{CRDS}$ interface, we also noticed an additional contact between EDA·A1$_{THD}$ with EDAR$_{CRDS}$ from the adjacent asymmetric unit in the crystal structure (Supplementary Fig. 2d, e), in which Ser275 and Arg276 of EDA·A1$_{THD}$ interact with the Glu55, Pro73, and Ala76 of the EDAR$_{CRDS}$ (Supplementary Fig. 2f). Gel filtration chromatography analysis using purified proteins revealed a molecular weight of about 100 kD for the EDA·A1$_{THD}$-EDAR$_{CRDS}$ hexameric complex, showing no sign of higher-order EDA·A1$_{THD}$-EDAR$_{CRDS}$ assembly formation (Supplementary Fig. 1a). This observation suggests that the EDA·A1$_{THD}$-EDAR$_{CRDS}$ contact between adjacent asymmetric unit cells either is very weak in solution or it is formed only because of the lattice packing effect in the crystal.

**Structure basis for ligand-receptor specificity of EDA·A1**
EDA·A1 and EDA·A2 specifically interact with their cognate receptors EDAR and XEDAR respectively, playing different roles in skin appendage development[10,18]. Notably, EDA·A1 only differs from EDA·A2 by two extra amino acids Val307 and Glu308 in EDA·A1. It has been an intriguing question how this subtle difference between EDA·A1 and EDA·A2 determines the specificity for their respective receptors[10–12].

The crystal structure of the EDA·A1$_{THD}$-EDAR$_{CRDS}$ complex provided us with a unique opportunity to address this issue. Surprisingly, the structure unveils that, although Val307 and Glu308 are in the vicinity of the EDA·A1$_{THD}$-EDAR$_{CRDS}$ interface, neither of them directly contributes to the interaction with EDAR (Fig. 3a). Instead, the presence of Val307 and Glu308 extend the length of strand βC of EDA·A1$_{THD}$ so that the aromatic sidechains of Tyr310, Try311 and Tyr343 pack together to form a hydrophobic patch that makes intimate contacts with helix α1 from EDAR$_{CDR2}$ (Fig. 2f). Superposition of the EDA·A2$_{THD}$ structure onto that of the EDA·A1$_{THD}$-EDAR$_{CRDS}$ complex reveals that, in contrast, the lack of Val307 and Glu308 markedly reshapes the local geometry of EDA·A2$_{THD}$ at the vicinity of the interface (Fig. 3b). Positions equivalent to EDA·A1$_{THD}$ Tyr310 and Tyr311 are occupied by Ile312 and Asn313 respectively in EDA·A2$_{THD}$, losing the hydrophobic interactions with EDAR$_{CRDS}$ (Fig. 3b). Furthermore, the lack of Val307 and Glu308 in EDA·A2 causes Phe314 rotating from the contact with EDAR$_{CRDS}$ and being buried inside the β-sandwich in EDA·A2$_{THD}$ (Fig. 3c). The buried aromatic side chain of Phe314 in EDA·A2 alters the position and orientation of loop EF (Fig. 3c), thus disrupting the interaction between Lys340 and Thr341 in loop EF and Glu94$^{EDAR}$ observed in the EDA·A1$_{THD}$-EDAR$_{CRDS}$ complex (Fig. 3b). Together, this comparative structural analysis shows that the two extra amino acids in EDA·A1 defines the structural basis for the specific interaction between EDA·A1 and EDAR.

To further investigate the receptor specificity, we overlaid the AlphaFold-predicted structure of XEDAR$_{CRDS}$ (residue 17–105) onto the EDA·A1$_{THD}$-EDAR$_{CRDS}$ complex (Fig. 3d and Supplementary Fig. 2g, h)[19]. We found that several structural features could explain why EDA·A1$_{THD}$ is incompatible with XEDAR$_{CRDS}$. First, replacing EDAR Gly95 by XEDAR Val66 induces a steric hindrance between XEDAR Val66 and EDA·A1 Tyr310 (Fig. 3d). Second, the substitution of EDAR Asp92 with XEDAR Thr63 abolishes the interaction with EDA·A1 Tyr310 (Fig. 3d). Third, the overall hydrophobic nature of helix α1 in XEDAR$_{CRD2}$ would

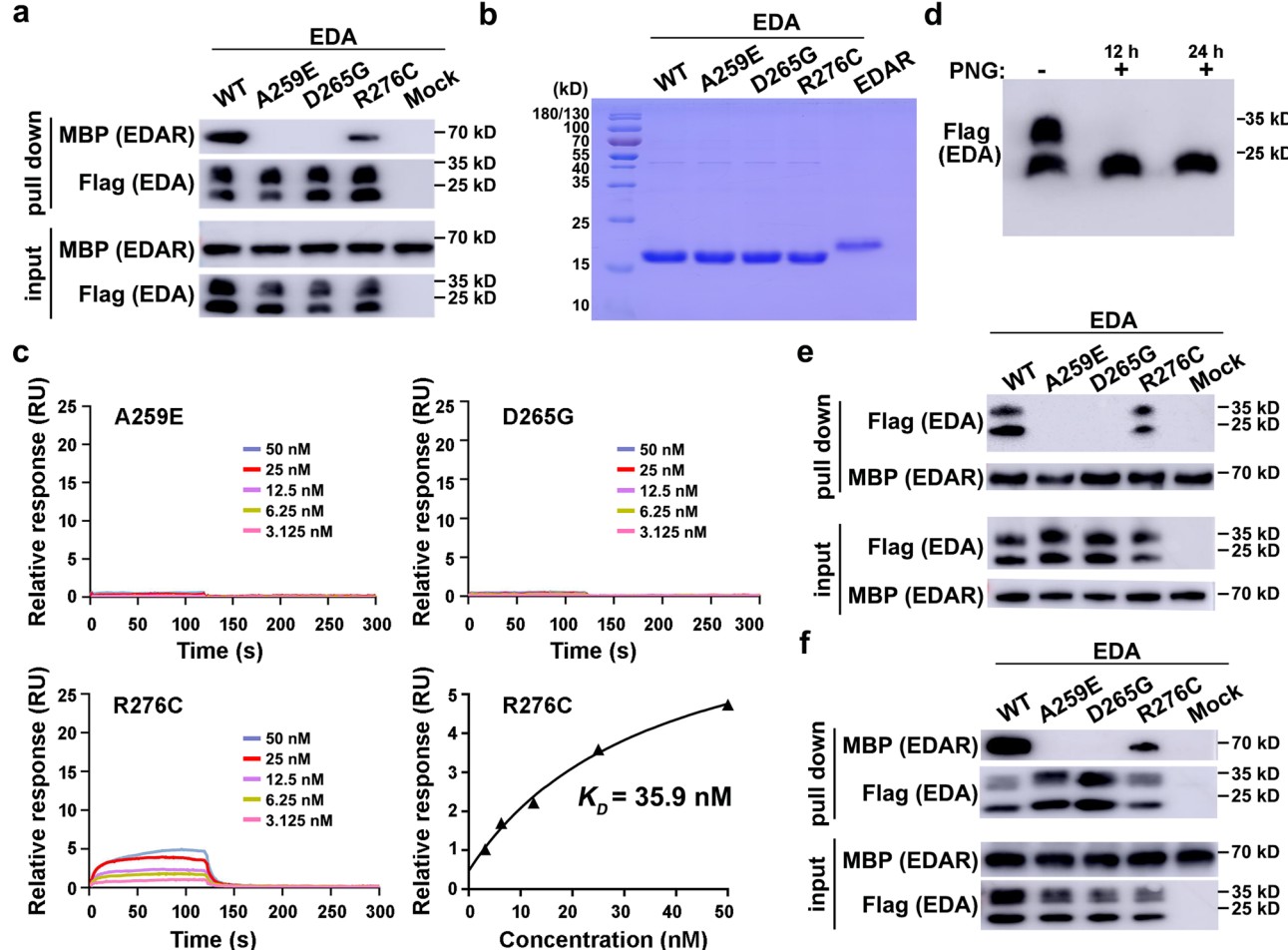

**Fig. 4 | Biochemical analysis of mutations at the EDA·A1$_{THD}$-EDAR$_{CRDS}$ interface. a** Pull-down assay using anti-Flag beads with ectopically expressed human Flag-EDA·A1$_{THD}$ and purified EDAR$_{CRDS}$-MBP. The levels of each protein in the input and pull-down samples were analyzed by immunoblotting with the indicated antibodies. The upper and lower bands of Flag-EDA·A1$_{THD}$ are N-glycosylated and non-N-glycosylated isoforms, respectively. WT, wild type; MBP (EDAR), anti-MBP to detect EDAR$_{CRDS}$-MBP; Flag (EDA), anti-Flag to detect Flag-EDA·A1$_{THD}$. **b** SDS-PAGE analysis of EDA·A1$_{THD}$ variants and EDAR$_{CRDS}$. **c** Surface plasmon resonance measurement of binding affinity between EDA·A1$_{THD}$ variants and EDAR$_{CRDS}$. **d** Deglycosylation of EDA·A1$_{THD}$. Soluble Flag-EDA·A1$_{THD}$ proteins expressed in the supernatant of HEK293T cells were treated with or without peptide N-glycanase

(PNGase) F for 12 hours (12 h) or 24 hours (24 h) before western blotting analysis. N-glycosylated forms of Flag-EDA·A1$_{THD}$ (upper bands) disappeared after being treated with PNGase F. **e** Pull-down assay using amylose (MBP-tag affinity) agarose beads with ectopically expressed human Flag-EDA·A1$_{THD}$ and purified EDAR$_{CRDS}$-MBP. The levels of each protein in the input and pull-down samples were analyzed by immunoblotting with the indicated antibodies. WT, wild type; MBP (EDAR), anti-MBP to detect EDAR$_{CRDS}$-MBP; Flag (EDA), anti-Flag to detect Flag-EDA·A1$_{THD}$. **f** Pull-down assay using anti-Flag beads with ectopically expressed mouse Flag-EDA·A1$_{THD}$ and purified mouse EDAR$_{CRDS}$-MBP. The levels of each protein in the input and pull-down samples were analyzed by immunoblotting with the indicated antibodies. Source data are provided as a Source Data file.

lead to the loss of the hydrogen-bonding interactions observed in the EDA·A1$_{THD}$-EDAR$_{CRDS}$ complex (Fig. 3d). Therefore, we conclude that the unique sequence of XEDAR is evolved for specific interaction with EDA·A2 but not with EDA·A1. How XEDAR specifically recognizes EDA·A2 awaits future investigations.

**Mutational analysis of the EDA·A1-EDAR interface**
To corroborate our structural analysis, we examined whether missense mutations at the interface could weaken or disrupt the interaction between EDA·A1$_{THD}$ and EDAR$_{CRDS}$. In particular, we focused on two disease-causing mutations in EDA·A1$_{THD}$ at the center of the interface, A259E and D265G, which clinically are NSTA and XL-HED causing pathogenic factors in humans respectively[20,21]. Both Ala259 and Asp265 are key residues on the middle acidic patch of EDA·A1$_{THD}$ essential for EDAR$_{CRDS}$ binding (Fig. 2e). Our crystal structure predicts that a glutamate substitution of Ala259 would severely deform the saddle-shaped depression of EDA·A1 whereas a glycine replacement of Asp265 reduces the acidic surface area at the interface (Fig. 2e). Both of these changes should interfere with the interaction between EDA·A1$_{THD}$ and

EDAR$_{CRDS}$. Consistent with this structural analysis, both A259E and D265G mutations in EDA·A1$_{THD}$ completely abolished the interaction with EDAR$_{CRDS}$ as revealed by a pull-down assay, underscoring the importance of A259$^{EDA·A1}$ and Asp265$^{EDA·A1}$ in the interaction with EDAR (Fig. 4a). This was also confirmed by an SPR assay using purified ectodomains of EDA·A1 and EDAR, which showed that both A259E and D265G mutant EDA·A1$_{THD}$ proteins exhibited no detectable interaction with EDAR$_{CRDS}$ (Fig. 4b, c). We noticed that the EDA·A1ectodomain purified from human embryonic kidney 293 T (HEK293T) showed two close bands in western blots (Fig. 4a). The higher molecular-weight band corresponded to a glycosylated form of EDA·A1$_{THD}$ as revealed by a deglycosylation analysis with peptide N-glycanase (PNGase) F (Fig. 4d). The effect of mutations interrupting the binding of EDAR was the same for both forms of EDA·A1$_{THD}$ (Fig. 4a, e), and similar results were obtained with mouse EDA·A1$_{THD}$ and EDAR$_{CRDS}$ (Fig. 4f).

To further investigate the effect of *EDA* mutations on the EDA·A1-EDAR interaction, we ectopically expressed full-length EDAR and wild-type (WT) or mutant ectodomains of EDA·A1 in human embryonic kidney 293T (HEK293T) cells. Immunofluorescence (IF) staining

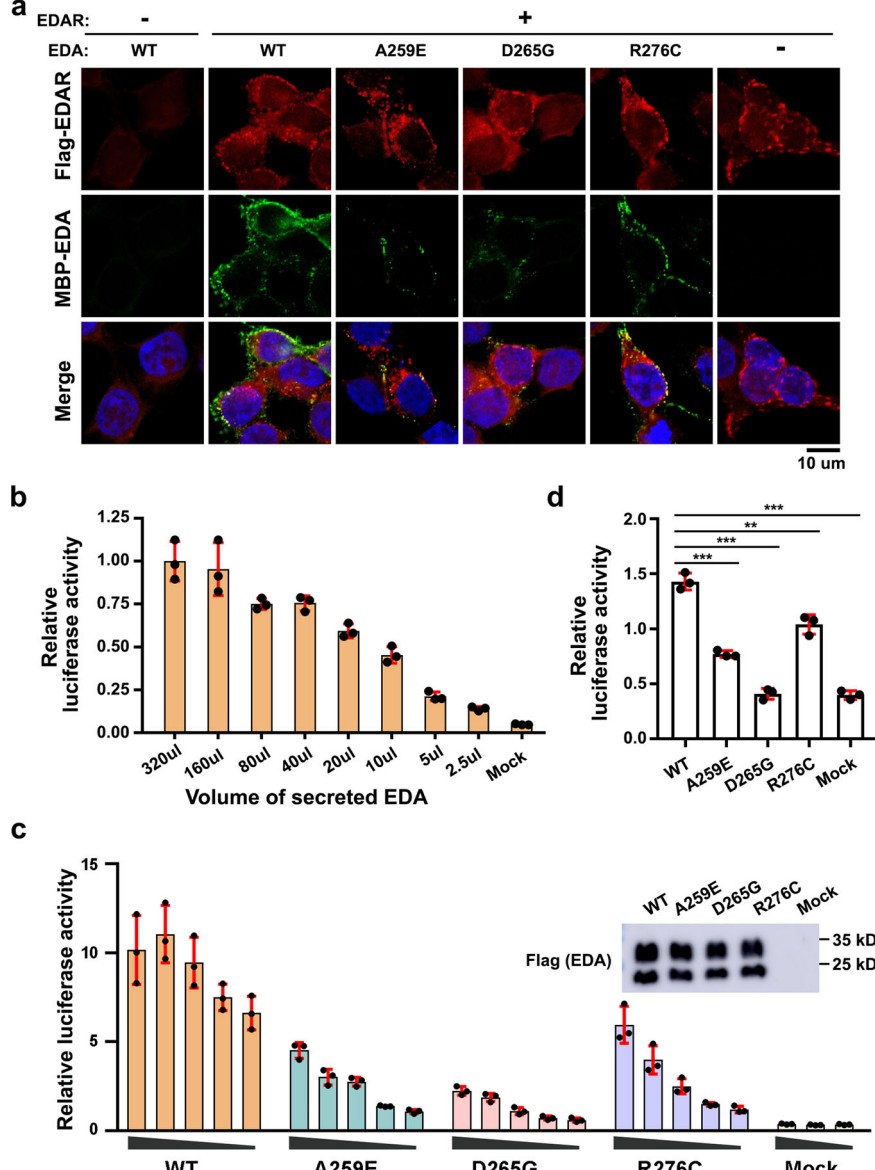

**Fig. 5 | Functional analysis of the EDA·A1-EDAR interaction mutations.**
**a** Immunofluorescence staining showing the binding of EDA·A1$_{THD}$ variants to EDAR on the cell membrane. MBP-EDA·A1$_{THD}$, full-length EDAR-Flag and DNA were stained in green, red and blue, respectively. Scale bar: 10 μm. **b** Luciferase reporter assay shows that supplemented WT EDA·A1$_{THD}$ activates NF-κB signaling pathway in a dose-dependent manner in HEK293T cells ectopically expressing full-length EDAR. Data are presented as the mean ± SEM for $n = 3$ independent experiments. **c** The interaction between EDAR$_{CRDS}$ and EDA·A1$_{THD}$ variants was assessed by a

luciferase reporter assay with HEK293T cells ectopically expressing full-length EDAR ($n = 3$, mean ± SEM). Top right: similar amount of WT and mutant soluble EDA·A1$_{THD}$ proteins from supernatants of transfected HEK293T cells were analyzed by Western blot to validate the amounts of proteins. **d** Luciferase reporter assay with HaCaT cells that express endogenous EDAR ($n = 3$, mean ± SEM). A two-sided Student $t$-test was performed. ***$P = 0.00016$ (A259E), ***$P = 0.000043$ (D265G), **$P = 0.0045$ (R276C) and ***$P = 0.000033$ (Mock). Source data are provided as a Source Data file.

analysis revealed that WT EDA·A1$_{THD}$ could be efficiently recruited to the plasma membrane in EDAR-expressing cells (Fig. 5a). By contrast, no EDA·A1$_{THD}$ signal was observed at the control HEK293T cell membrane, indicating that the appearance of cell membrane-attached EDA·A1$_{THD}$ is mediated by the interaction with ectopically expressed EDAR (Fig. 5a). In accordance with our structural and biochemical results, IF data clearly showed that mutations A259E or D265G almost depleted the signals of membrane-associated EDA·A1$_{THD}$ (Fig. 5a), suggesting that the EDA·A1-EDAR interaction observed in the crystal structure is essential for cell membrane clustering of EDA·A1 and EDAR.

Binding of EDA·A1 to EDAR during ectodermal organ placode formation leads to the activation of transcription factor NF-κB[22]. To further investigate the effect of EDAR-binding deficient mutations of

EDA·A1 in this process, we performed a luciferase reporter assay in HEK293T cells which mimics the EDA·A1-EDAR interaction-mediated activation of NF-κB[23,24]. As a control, we demonstrated that supplementation of purified WT EDA·A1$_{THD}$ could robustly activate the luciferase reporter transcription in a dose-dependent manner in HEK293T cells ectopically expressing EDAR (Fig. 5b). In contrast, supplementation of either A259E or D265G mutant EDA·A1 greatly impaired the transcriptional activation of the reporter (Fig. 5c). The mutation effect of EDA·A1 in driving NF-κB transcriptional activation was further confirmed using HaCaT cells that express endogenous EDAR (Fig. 5d). Notably, the defect of the D265G mutant was more prominent than that of A259E, suggestive of a more severe defect of the D265 mutant in transcriptional activation (Fig. 5c)[20,21]. These data also indicate that, compared to the in vitro biochemical and IF staining

analyses, the reporter assay is a more sensitive technique for evaluating the cellular defect of EDA·A1-EDAR interacting mutations (Figs. 4a, c and 5a, c). Taken together, our results demonstrate the essential role of the EDA·A1-EDAR interaction in the signal pathway of NF-κB transcriptional activation.

In addition to A259E and D265G, we also examined the defect of another disease-causing mutation R276C in EDA·A1. Patients with this mutation exhibit a milder XL-HED clinical phenotype than those of A259E and D265G patients[20,21,25]. Consistently, both in vitro pull-down analysis, SPR measurement, and IF staining data displayed only marginal differences between the WT and R276C mutant experiments (Figs. 4a, c and 5a, c). Further luciferase reporter assay unveiled a clear defect in transcriptional activation but to a less extent compared to the A259E and D265G mutations (Fig. 5c). It is noteworthy that EDA·A1 Arg276 coordinates an interaction with an EDAR molecule from the adjacent asymmetric unit in the crystal structure (Supplementary Fig. 2d–f). This interaction may play a role in mediating the interaction between EDA·A1 and EDAR, but is too weak to detect in in vitro biochemical assays (Figs. 4a, c and 5a). Nevertheless, both the identification of the R276C mutation in XL-HED patients and the defect it caused in the transcriptional reporter assay indicate that this mutation likely interferes with the in vivo interaction between EDA·A1 and EDAR and subsequent NF-κB activation, ultimately leading to the XL-HED phenotype (Figs. 4a, c and 5a, c).

### EDA mutations affect ectodermal development in mice
EDA pathway is conserved in most vertebrates[26]. Concordantly, multi-sequence alignment of EDA orthologs from xenopus to humans unveils a high-sequence identity among their ectodomains (Supplementary Fig. 3a). To further evaluate the in vivo function of the EDA·A1-EDAR interaction, we set out to generate knock-in mice with diseasing-causing mutations A259E, D265G, and R276C in EDA·A1 using the CRISPR-Cas9 method (Supplementary Fig. 3b)[27]. Stable lines were generated and male hemizygotes were subjected to biochemical and genetic investigations. A mutant mouse line that contains a frameshift insertion at Asn157 was used as an Eda knockout control (Eda^{ko/Y}) (Supplementary Fig. 3b). Although single-site Eda mutant and the Eda knockout mice were indistinguishable from their WT littermates in developmental growth and viability, these mutant mice exhibited varying degrees of developmental defects in their ectodermal derivatives including hair, teeth, and sweat glands.

The defects in Eda^{ko/Y} mice were the most severe and characterized by kinked tail tips, hairless tails and abdomen, a bald patch behind ears, as well as abnormal eyelid development (Fig. 6a). The Eda^{ko/Y} mice had only two mandibular molars with the third molar missing (Fig. 6b). The lower first molar (M1) was small with only three rounded cusps substantially different from those of WT mice, whose crown is usually made of seven deep and well-defined cusps linked by transverse crests (Fig. 6b). The lower second molar (M2) of Eda^{ko/Y} mice exhibited a high incidence of taurodontism with a very large pulp chamber (Fig. 6b, c). In addition to dental agenesis, Eda^{ko/Y} mice also displayed a severe defect in the sweating ability as revealed by the Starch-iodine test as well as the histological staining analysis that showed no eccrine sweat glands in the footpads of Eda^{ko/Y} mice (Fig. 7a–c). Furthermore, histological analysis also uncovered an evident decrease in the density of hair follicles in the abdominal skin from the Eda^{ko/Y} mice (Fig. 7d), indicating that hair development is affected by Eda knock-out in mice. These defects of the Eda^{ko/Y} mice are consistent with those of previously reported tabby mice that harbored a null allele of the Eda gene[28,29].

Among the single-site missense mutant mice, the Eda^{D265G/Y} mice displayed the most severe defects in ectodermal organ development including scanty abdomen hair and eyelid (Fig. 6a). Although all three mandibular molars were present in the Eda^{D26SG/Y} mutant mice, they displayed reduced and rounded cusps (Fig. 6b); their lower M1 were

smaller with only four of normal seven cusps formed, and their lower M2 also exhibited a high incidence of taurodontism (Fig. 6b, c), similar to that in the Eda^{ko/Y} mice (Fig. 6b, c). Investigation of the sweating abilities found that the Eda^{D265G/Y} mice resembled the knock-out mutant, showing almost no sweating function and very few sweat glands observed in the footpads (Fig. 7a–c). Further examination of abdominal skin tissue sections of the Eda^{D265G/Y} mice revealed an obvious decrease in the number of hair follicles, which is in alignment with the observed scanty abdomen hair (Figs. 6a and 7d). Collectively, these data suggest that the Eda^{D265G/Y} mutation causes ectodermal organ developmental defects similar to those of the Eda knockout mice but with a slightly milder phenotype.

Compared to the D265G mutant, the A259E and R276C mutations only caused mild defective phenotypes in mice with no obvious abnormalities in abdomen hair, eyelid and tail hair and tips (Fig. 6a). The number and the size of mandibular molars were normal in Eda^{A259E/Y} and Eda^{R276C/Y} mice, but the molars still displayed more flattened, rounded cusps different from the deep and defined cusps of the WT molars (Fig. 6b). No obvious taurodontism malformations were observed in lower M2 in Eda^{A259E/Y} and Eda^{R276C/Y} mice (Fig. 6b, c). The sweating function and the formation of sweat glands in these two mutant mice exhibited an intermediate level of defects compared to the Eda knock-out and Eda^{D265G/Y} mice (Fig. 7a–c). Closer inspection unveiled that the sweating defects of Eda^{A259E/Y} were consistently more severe than those of Eda^{R276C/Y} (Fig. 7a–c). Further histological examination showed no significant decrease in the number of hair follicles in Eda^{A259E/Y} and Eda^{R276C/Y} mice (Fig. 7d).

Although skull and facial dysmorphologies have been reported in human patients suffering from HED[20,30], we observed no substantial craniomaxillofacial deformities in Eda knock-out mice (Fig. 7e). This result indicates that craniomaxillofacial development in different species depends on the EDA function in varying degrees and mouse craniomaxillofacial development exhibits less vulnerable to the deficiency in EDA signaling. The EDA gene is located on the X chromosome and male carriers of EDA variants (hemizygous for an EDA mutation) tend to be more affected than heterozygous female carriers who may have mild-to-asymptomatic manifestations due to somatic mosaicism[2]. We also analyzed homozygous Eda mutant female mice and observed similar HED phenotypes to those in hemizygous male mice (Supplementary Fig. 4), suggesting that EDA mutations affect ectodermal development in both male and female mice.

Next, we compared the developmental defects in ectodermal derivatives of these mutant mice with the effects of the corresponding mutations in human patients (Fig. 7f)[20,21,25]. A patient carrying the D265G^{EDA} mutation was diagnosed with severe XL-HED, characterized by facial abnormalities in addition to the typical triad of oligodontia, hypohidrosis and hypotrichosis (Fig. 7f)[20]. In contrast, patients with the A259E^{EDA} or R276C^{EDA} mutations exhibited mild clinical manifestations with fewer missing teeth (15 to 16 versus ~23 in XL-HED patients) and normal shape and size in the remaining teeth (Fig. 7f)[21,25]. Moreover, these affected patients also had normal sweating and fine scalp, except that the R276C^{EDA} patients display somewhat sparse eyebrows (Fig. 7f)[21,25]. Taken together, the phenotype severities of diseasing-causing EDA mutations revealed by the mouse model are consistent with not only the in vitro biochemical and cellular assays but also the reported clinical patient phenotypes.

## Discussion
Defects in the EDA gene have long been known to be linked with XL-HED, the most frequent form of HED that has been documented for more than 140 years[13]. Despite that a series of EDA mutations have been identified in XL-HED and NSTA patients, the underlying molecular mechanism of how EDA mutations give rise to different clinical phenotypes remains largely unknown. In this study, we determine the structure of the EDA·A1_{THD}-EDAR_{CRDS} complex, reveal the atomic

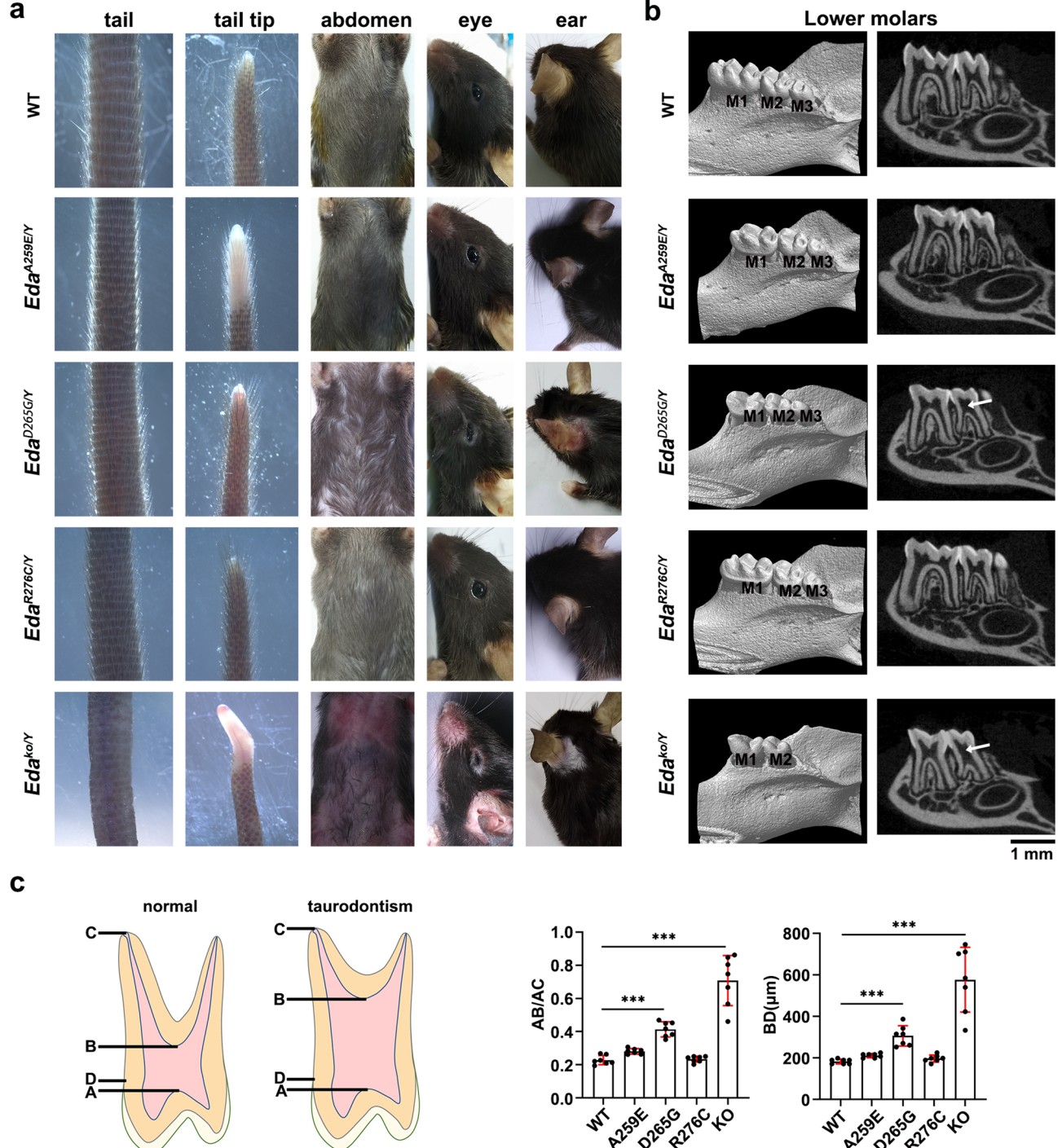

**Fig. 6 | Disruption of the EDA·A1$_{THD}$-EDAR$_{CRDS}$ interaction results in ecto-dermal dysplasia in mice. a** The *Eda* mutant male mice were analyzed for their ectodermal derivatives. The *Eda*$^{ko/Y}$ mice showed the most severe defects characterized by hairless tails and abdomen, kinked tail tips, a bald patch behind ears and abnormal eyelid development. The phenotype of the *Eda*$^{D265G/Y}$ mice was slightly milder than that of the *Eda*$^{ko/Y}$ mice, showing scanty abdomen hair and ear hair, but with normal tail hair and tips. The *Eda*$^{A259E/Y}$ and *Eda*$^{R276C/Y}$ mutant mice exhibited no obvious abnormalities in the ectodermal derivatives mentioned above. **b** Representative radiographic images of lower molars from WT and mutant *Eda* mice. White arrows indicate "bull-shaped" taurodontism teeth with a large pulp

cavity. M1, M2, M3: first, second, third molar. Scale bar: 1 mm. **c** Quantification of the taurodontism phenotype in *Eda* mutant mice. Left: Schematic diagram of the taurodontism phenotype. Taurodontism is characterized by an elongation of the pulp chamber extending into the root area. A, pulp roof; B, pulp floor; C, apex of the longest tooth root; D, enamel-cemental junction. Right: Quantification of taur-odontism phenotype. Data are presented as the mean ± SD for $n = 7$ adult (about 6-week old) male mice per group. A two-sided Student's *t*-test was performed. ***$P = 8.7E{-}07$ (AB/AC, D265G), ***$P = 2.7E{-}06$ (AB/AC, KO), ***$P = 2.8E{-}05$ (BD, D265G) and ***$P = 2.4E{-}05$ (BD, KO). Source data are provided as a Source Data file.

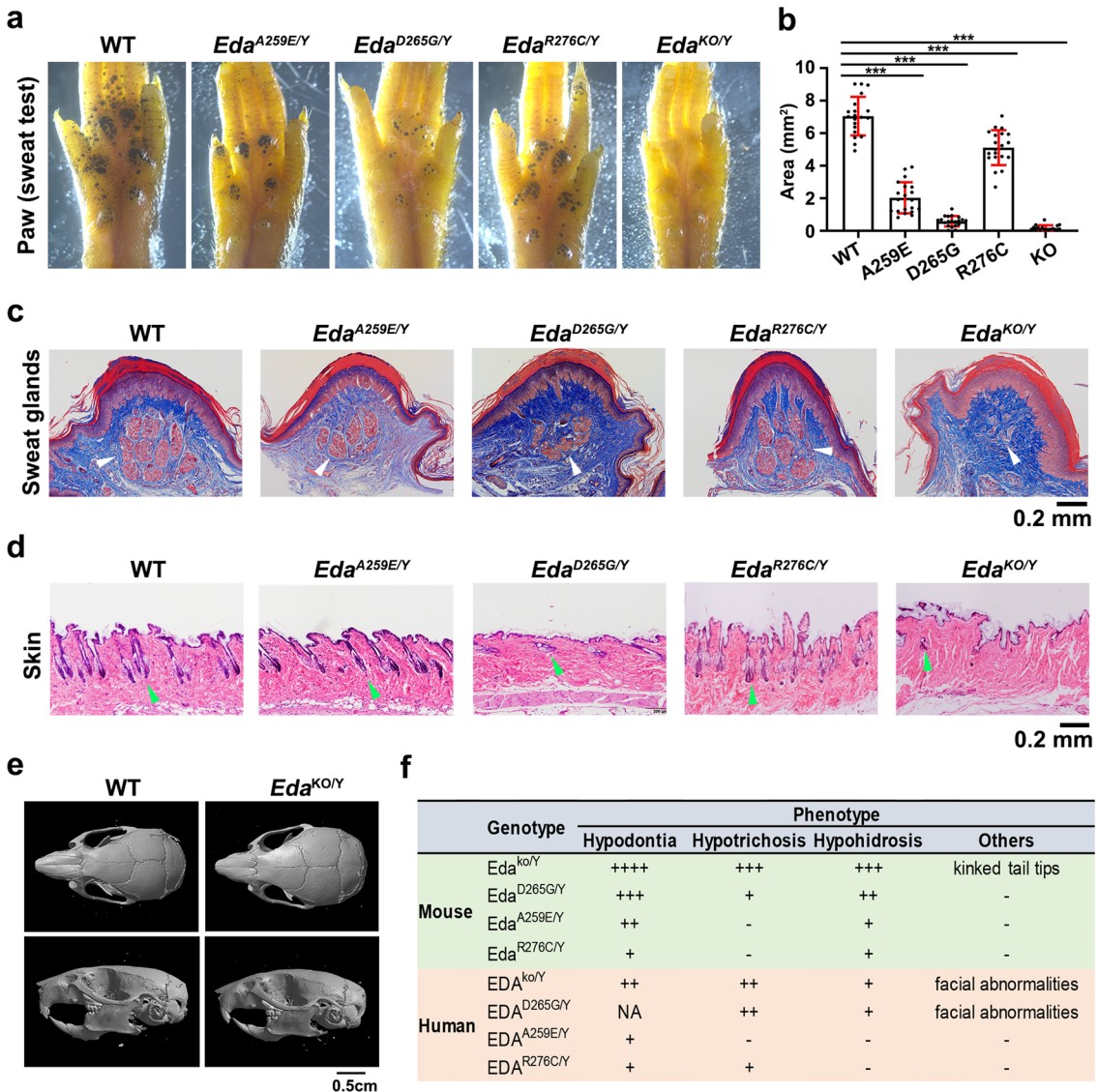

**Fig. 7 | Sweat gland and hair follicle phenotypes in EDA-EDAR interaction deficiency mice. a** Sweat test of WT and mutant *Eda* mice. Sweat is detected as dark spots. **b** The area of sweat dark spots from **a** is quantified as an indication of the sweating ability. Data are presented as the mean ± SD for n = 9 male mice for KO and 10 for other genotypes with measurements recorded from the two hind paws of each individual. A two-sided Student's *t*-test was performed. ***P = 2.2E−04 (A259E), ***P = 3.1E−24 (D265G), ***P = 3.7E−06 (R276C) and ***P = 6.3E−24 (KO). **c** Histological sections of the footpads. Sweat glands are indicated by white arrowheads. Scale bar: 0.2 mm. **d** Histological sections of the abdominal skin showing hair follicles indicated by green arrowheads. Scale bar: 0.2 mm. **e** Micro−computed tomography (μCT) analysis of craniofacial phenotypes in *Eda* knockout and WT mice. No obvious craniomaxillofacial abnormality was observed in *Eda* knockout mice. **f** Phenotypic variations in mice and humans carrying EDA

mutations. +, ++, and +++: degree of severity; - within the normal range; NA: not available. Hypodontia score (mouse): -, normal molars; +, reduced and rounded cusps; ++, reduced and rounded cusps, missing cusps; +++, reduced and rounded cusps, missing cusps, taurodontism; ++++, reduced and rounded cusps, missing cusps, taurodontism, missing the third molar. Hypodontia score (human): NA, not available; +, missing teeth; ++, missing teeth, peg-shaped residual teeth. Hypotrichosis score (mouse): -, dense abdomen hair; +, sparse abdomen hairs; ++, very few abdomen hairs; +++, no abdomen hair. Hypotrichosis score (human): -, dense hair; +, sparse eyebrows; +, very few eyebrows, sparse body and scalp hair. Hypohidrosis score (mouse): +, slightly reduced sweat volume; ++, severe reduced sweat volume; +++, no sweat volume. Sweating function score (human): -, able to sweat; +, unable to sweat. Source data are provided as a Source Data file.

details of the ligand-receptor interaction, and examine the deleterious effects of *EDA* mutations in mouse models. Our structural data show that three disease-causing mutations, D265G, A259E, and R276C in *EDA*[20,21,25], are all located at the ligand-receptor interface (Fig. 2). Further functional studies reveal a varying degree of EDA deficiency in both the interaction with EDAR and the activation of the downstream NF-κb signaling pathway, consistent with the observed defect severities in the ectodermal development in mouse models and human patients (Figs. 2, 5c, 6 and 7a–d). These results support the idea that the interface mutations that completely disrupt the EDA-EDAR interaction leads to severe XL-HED phenotypes, while those that only

partially weaken the interaction result in mild non-syndromic manifestations.

So far there are about 100 pathogenic missense mutations in EDA·A1$_{THD}$ registered in HGMD (Supplementary Table 2) (http://www.hgmd.cf.ac.uk/). Based on our structural analysis of the EDA·A1$_{THD}$-EDAR$_{CRDS}$ complex, these disease-causing mutations could be divided into four distinct groups: mutations in the interior of EDA·A1$_{THD}$ that lead to protein destabilization (type I); mutations at the protomer interface which affect the trimer formation of EDA·A1 (type II); mutations at the ligand-receptor interface which interfere with the interaction with EDAR (type III); and mutations located at the EDA exterior

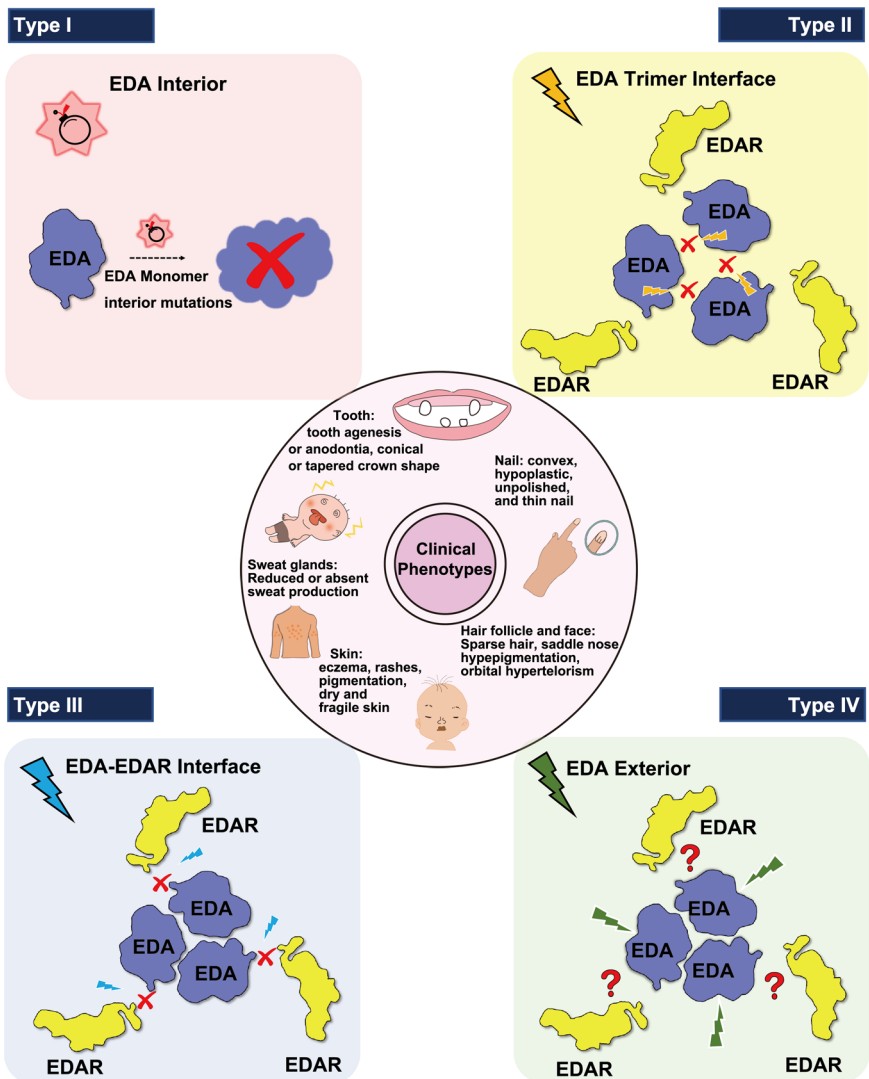

**Fig. 8 | Summary of missense mutations in EDA·A1$_{THD}$.** The pathogenic mutations that cause ectodermal dysplasia (middle circle) can be divided into four distinct categories (type I, II, III, and IV) with different locations in EDA·A1$_{THD}$ and distinct clinical phenotypes.

region away from the EDAR binding interface (type IV) (Fig. 8). The majority of residues corresponding to type-I mutations have side chains buried inside the β-sandwich of EDA·A1, substitution of which likely affects protein folding via changing local hydrophobicity or introducing a bulky group or secondary structure disruption, ultimately leading to the inactivation of the entire protein (Fig. 8). Thus, affected patients carrying EDA·A1 type I mutations mostly exhibit severe clinical manifestations of XL-HED phenotypes (Fig. 8 and Supplementary Table 2). Similarly, among type II mutations that affect the trimeric interaction of EDA·A1, we also observe a high percentage of mutations accounting for severe phenotypes of ectodermal dysplasia syndromes (Fig. 8 and Supplementary Table 2), suggestive of a key role of the trimeric structure of EDA in the development. Although the EDA type III mutations impact neither the EDA folding nor the trimer formation, substitution of key residues at the ligand-receptor interface also leads to ectodermal dysplasia manifestations, indicating that the EDA·A1-EDAR interaction is essential to the ectodermal organ development (Fig. 8 and Supplementary Table 2). Type IV EDA·A1 mutations include apparently innocuous point mutations located at the exterior of EDA·A1 away from the EDA·A1-EDAR interface (Fig. 8). Structural information of the EDA·A1$_{THD}$-EDAR$_{CRDS}$ complex could not explain the developmental defects caused by these mutations (Fig. 8). Further

functional studies of these mutations are required to reveal their pathogenicity.

In this work, our structural, biochemical, and cellular analyses unveil the structural basis of the specific interaction between EDA·A1 and its cognate receptor EDAR, revealing an essential role of the EDA·A1-EDAR interaction in the signal pathway of NF-κB transcriptional activation. Moreover, our studies on the structure-guided knock-in mouse models demonstrate that different EDA·A1 mutations lead to varying degrees of developmental defects in mouse ectodermal derivatives, which is consistent with the clinical observations on patients. In summary, our work provides important insight into the EDA signaling mechanism and the structural basis for the systematic study of disease-causing mutations in *EDA* in humans.

## Methods

### Genes, plasmids

The optimized cDNA encoding the THD domain of EDA·A1 (residues 233–391) was cloned into the pMAL-c2x vector (Addgene) with an N-terminal MBP (maltose binding protein)-tag (primers: EDA-opt-233-391-F and EDA-opt-233-391-R, Supplementary Table 3). The sequence encoding the ectodomains of EDAR (EDAR$_{CRDS}$, residues 30–150) was cloned into the baculovirus transfer vector pFastBac (Invitrogen),

in-frame with an N-terminal EDAR (residues 1–28) signal peptide for secretion, an MBP-tag for purification, an N-terminal 3 C cleavage site and a linker sequence (GGSGGSGGSGGS) (primers: EDAR-30-150-F and EDAR-30-150-R, Supplementary Table 3). Secreted EDA·A1 THD domain was cloned into the expression vector pcDNA3.1 (Addgene), containing an HA signal peptide, a Flag tag, a linker (GGSGGSGGSGGS), and amino acids 233–391 of EDA·A1 (primers: EDA-233-391-F and EDA-233-391-R, Supplementary Table 3). The cDNA of full-length EDAR was cloned into the pLVX-puro vector (Addgene) with a C-terminal Flag tag (primers: EDAR-1-448-F and EDAR-1-448-R, Supplementary Table 3). All mutants were generated by the site-directed mutagenesis kit (Agilent) with primers listed in Supplementary Table 3 and verified by DNA sequencing.

## Cell culture and transient transfection
Human embryonic kidney 293T cells (HEK293T) and Human Keratinocytes cells (HaCaT) were purchased from the Type Culture Collection of the Chinese Academy of Sciences, Shanghai, China. Cells were cultured in Dulbecco's modified Eagle's medium (Gibco) supplemented with 10% fetal bovine serum (Gibco) at 37 °C under 5% $CO_2$. Plasmid transfection was performed using Lipofectamine 3000 (Invitrogen) according to the manufacturer's instructions.

## Animals
Adult WT C57BL/6J mice and ICR (CD-1) mice were purchased from Shanghai SLAC laboratory animal Co. Ltd. Mice were housed under constant ambient temperature ($22 \pm 2$ °C) and humidity ($55 \pm 10\%$), with an alternating 12-h light/dark cycle. Water and food were available ad libitum. Experimental protocols were approved by the Institutional Animal Care and Research Advisory Committee of the Shanghai Ninth People's Hospital, School of Medicine, Shanghai Jiao Tong University (approval # SH9H-2019-A130-1). Every effort was made to minimize and refine the experiments to avoid animal suffering.

## Antibodies
The following antibodies were used in this study: secondary antibodies for IF and western blotting: Alexa Fluor 594 Goat anti-Rabbit IgG (Invitrogen, A-11012, 1:500, RRID: AB_2534079), Alexa Fluor 488 Goat anti-Mouse IgG (Invitrogen, A-11001, 1:500, RRID: AB_2534069), HRP-conjugated Goat anti-Mouse IgG (Proteintech, SA00001-1, 1:4000, RRID: AB_2722565), and HRP-conjugated Goat anti-Rabbit IgG (Proteintech, SA00001-2, 1:4000, RRID: AB_2722564). Rabbit antibodies against FLAG (Proteintech, 20543-1, 1:2000, RRID: AB_11232216). Mouse antibodies against MBP (Proteintech, 66003-1, 1:2000, RRID: AB_11183040).

## Protein expression and purification
*E. coli* BL21 CodonPlus (DE3) cells (Stratagene) were transformed with a pMAL-c2x vector expressing EDA·A1$_{THD}$. Transformed cells were grown at 37 °C, cooled at 4 °C for 30 min when OD reached 0.6, and protein expression was induced for 18 hours with 1 mM IPTG at 16 °C. Cells were harvested by centrifugation, and resuspended in lysis buffer (25 mM MES, pH 6.5, 500 mM NaCl, 10% glycerol, 1 mM PMSF, 5 mM benzamidine, 1 μg/ml leupeptin and 1 μg/ml pepstatin), and lysed by sonication. The cell lysate was ultracentrifuged and the supernatant was incubated with amylose (MBP-tag affinity) agarose beads (NEB) at 4 °C with rocking for 3 h. After extensive washing, the bound proteins were eluted by 25 mM maltose. Eluted proteins were then concentrated and purified using a HiLoad 16/600 Superdex 200 PG column (GE Healthcare) equilibrated with the buffer (25 mM MES, pH 6.5, 200 mM NaCl). Protein samples were subjected to 3 C protease (BD Biosciences, 3 units/mg proteins, overnight at 4 °C) to remove the N-terminal MBP tag, followed by ion-exchange chromatography using Mono-S 4.6/100 PE column (Sigma-Aldrich) by gradient elution with 0 to 1 M NaCl (25 mM MES, pH 6.5). The fractions containing EDA·A1$_{THD}$

were further purified on a Superdex 200 increase 10/300 GL column (GE Healthcare) equilibrated in 25 mM MES (pH 6.5) and 200 mM NaCl. The method for the expression and purification of the mutated EDA·A1$_{THD}$ proteins was the same as that used with the WT EDA·A1$_{THD}$ proteins. The ectodomains of EDAR (EDAR$_{CRDS}$) were expressed in insect cells using the Bac-to-Bac baculovirus expression system (Invitrogen). Briefly, a pFastBac construct for secreted EDAR$_{CRDS}$ was used for transfection and virus amplification with Sf9 cells, and the EDAR$_{CRDS}$ proteins were produced by infecting suspension cultures of Hi5$^{TM}$ cells (Invitrogen) for 3 days. The culture supernatants were filtered through a 0.22-μm membrane and incubated with amylose agarose beads at 4 °C for 3 h, followed by extensive washing with 25 mM MES (pH 6.5), 200 mM NaCl. Proteins were eluted with 25 mM maltose, subjected to 3 C protease (BD Biosciences, 3 units/mg proteins) to remove the N-terminal MBP tag, and purified using a Mono-Q 4.6/100 PE column (Sigma-Aldrich) by gradient elution with 0–1 M NaCl (25 mM MES, pH 6.5). EDAR$_{CRDS}$ proteins were further purified by a Superdex-200 column equilibrated with the buffer (25 mM MES, pH 6.5, 200 mM NaCl). Proteins were concentrated and stored at −80 °C before use in crystallization, SPR analysis and pull-down assay.

## Crystallization and structure determination
Purified EDA·A1$_{THD}$ and EDAR$_{CRDS}$ were mixed at a molar ratio of 1:1, incubated on ice for 1 h, and concentrated to 16 mg/ml. Crystallization of the EDA·A1$_{THD}$-EDAR$_{CRDS}$ complex was screened by sitting-drop vapor diffusion at 4 °C. Crystals were grown in a reservoir solution of 25% (w/v) polyethylene glycol-6000, 0.1 M BICINE (pH 9.0) for about 2 weeks, and were cryo-protected by briefly soaking in reservoir solution supplemented with 25% (v/v) glycerol before flash-cooling in liquid nitrogen. Datasets were collected under cryogenic conditions (100 K) at the Shanghai Synchrotron Radiation Facility (SSRF) beamlines BL19U1. A 2.8-Å dataset of the EDA·A1$_{THD}$-EDAR$_{CRDS}$ complex was collected at the wavelength of 0.97853 Å, and the structure was determined by molecular replacement using the previously published EDA·A1$_{THD}$ structure (PDB: 1RJ7) as the searching model[11]. The atomic models were completed with Coot[31] and further structural refinement was carried out with Phenix[32]. Data collection, processing, and refinement statistics are summarized in Supplementary Table 1. All the crystal structural figures were generated using the PyMOL Molecular Graphics System (http://www.pymol.org).

## Surface plasmon resonance (SPR) analysis
SPR technology-based binding assays were performed using a Biacore 8 K instrument (GE Healthcare) with a running buffer (10 mM HEPES, pH 7.5, 150 mM NaCl, and 3 mM EDTA) at 25 °C. The WT and mutant EDA·A1$_{THD}$ proteins were diluted to 20 μg/ml in 10 mM sodium acetate buffer (pH5.5) and immobilized onto sensor CM5 chips (GE Healthcare) according to the manual. Purified EDAR$_{CRDS}$ proteins were serially diluted and injected into the sensor chips at a flow rate of 30 μl/min for 120 s (contact phase), followed by 180 s of buffer flow (dissociation phase). The $K_D$ value was derived using Biacore 8 K Evaluation Software Version 1.0 (GE Healthcare) and steady-state analysis of data at equilibrium.

## Western blotting
Cells were harvested and proteins were extracted in radio-immunoprecipitation assay (RIPA) buffer (Beyotime, P0013K) supplemented with complete Protease Inhibitor mixture (Roche). Cell lysates were centrifuged and supernatants were subjected to SDS-PAGE separation before transferred to PVDF membranes (GE Healthcare). The blots were incubated in blocking buffer (5% fat-free milk in PBS buffer supplemented with 0.05% TWEEN-20) at room temperature (RT) for 30 min and incubated with primary antibodies in blocking buffer at 4 °C overnight. Blots were then washed and incubated in the HRP-labeled secondary antibodies at RT for 1 h. After washing, blots

were developed with ECL Prime Western Blotting System (GE Healthcare, RPN2232).

## Expression of soluble EDA·A1$_{THD}$

HEK293T cells were seeded in a 12-well plate the day before transfection. Transfections were performed using 1 μg of vectors (pcDNA3.1) encoding WT or mutant soluble Flag-tagged EDA·A1$_{THD}$ protein and Lipofectamine 3000 (Invitrogen). Cells were cultured in DMEM supplemented with 10% fetal bovine serum (FBS). The supernatants were harvested 48 h after the transfection, and were subjected to western blotting, pull-down assay and luciferase assay.

## In vitro pull-down analysis

Human and mouse EDAR$_{CRDS}$-MBP proteins were expressed and purified from Hi5$^{TM}$ cells. WT or mutant human and mouse soluble Flag-tagged EDA·A1$_{THD}$ proteins were expressed in the supernatant of HEK293T cells. For pull-down assay, an equal volume of the supernatant of WT or mutated EDA·A1$_{THD}$ proteins was incubated with an equivalent amount of EDAR$_{CRDS}$-MBP proteins at 4 °C with shaking for 1 h before incubation with Flag beads (GE Healthcare) for 3 h. Beads were washed three times with binding buffer (25 mM Tris pH 7.5, 150 mM NaCl, 0.5% NP40) and subjected to western blotting to detect bound proteins.

## Generation of the EDAR-HEK293T stable cell line

HEK293T cells were seeded in a six-well plate before transfection and transfected with a pLVX-puro construct encoding full-length EDAR with a C-terminal Flag-tag. After 24 h, 1 ng/ml puromycin was used to select cells stably expressing EDAR protein for 7 days. The expression of EDAR was confirmed by western blotting and cells were subjected to EDA cell surface adsorption assay and luciferase assay.

## EDA cell surface adsorption assay

HEK293T cells stably expressing full-length EDAR were grown on coverslips (Thermo, T_7011254584) in a 12-well plate, and an equivalent amount of the purified WT or mutant MBP-EDA·A1$_{THD}$ proteins were added into each well. Cells were incubated with MBP-EDA·A1$_{THD}$ proteins at 37 °C for 1 h. After being washed three times with PBS, the cells were fixed with 4% paraformaldehyde for 20 min, permeabilized with 0.1% Triton X-100 in PBS and incubated with blocking buffer (PBS containing 0.1% Triton X-100 and 5% BSA) for 1 h at RT, and incubated with primary antibodies at 4 °C overnight. Coverslips with cells were then washed and incubated with fluorescence-conjugated secondary antibodies at RT for 1 h, thoroughly washed in PBS supplemented with 0.1% Triton X-100, air-dried and subjected to microscopy imaging by ZEISS, LSM 880.

## Luciferase assay

HEK293T cells stably expressing full-length EDAR or HaCaT cells were seeded in 12-well dishes one day before transfection. 500 ng of pNF-κB Luc plasmid (Promega, E8491), and 10 ng of pRL-TK Renilla reference plasmid (Promega, E2231) were co-transfected for each well using Lipofectamine 3000 (Invitrogen). Twenty-four hours after the transfection, an equivalent volume of the supernatant of WT or mutated EDA·A1$_{THD}$ proteins were added to each well, respectively. After 12 h (for HEK293T) or 18 h (for HaCaT), Firefly luciferase activity in the cell lysates was measured and normalized to Renilla luciferase activity using a dual-luciferase reporter assay system (Promega).

## Generation of mutant mouse lines using CRISPR/Cas9

*Eda* mutant mice were generated by CRISPR/Cas9-mediated genome engineering[27]. Guide RNA cassettes with sequences (gRNA-KO, gRNA-A259E, gRNA-D265G, gRNA-R276C) (Supplementary Table 3) were cloned into pX260 vector[33]. In vitro-transcribed Cas9 mRNA (100 ng/

μl) and guide RNA (50 ng/μl), and donor oligo (100 ng/μl, Donor-A259E, Donor-D265G, Donor-R276C) (Supplementary Table 3) were microinjected into the cytoplasm of fertilized eggs collected from C57BL/6J. The injected zygotes were cultured in M2 medium (Merck/Millipore) at 37 °C under 5% CO$_2$ overnight. The embryos that had reached two-cell stage of development were implanted into the oviducts of pseudo-pregnant ICR foster mothers. The mice born from the foster mothers were genotyped by PCR analysis of genomic tail-biopsy DNA (primers: mEDA-KO-F and mEDA-KO-R, mEDA-A259E-F and mEDA-A259E-R, mEDA-D265G/R276C-F and mEDA-D265G/R276C-R) (Supplementary Table 3). Founder mice were crossed to C57BL/6J mice to obtain germline transmission, and were backcrossed at least 3 generations to C57BL/6J to dilute any possible off-target effects of the CRISPR/Cas9 gene editing and to generate offspring for characterization.

## Hematoxylin-Eosin and Masson staining

Skin samples of the abdomen were collected from 6-week-old mice, fixed in 4% paraformaldehyde for 24 h, dehydrated in ascending series of alcohol, and then N-butanol, paraffin-embedded, and sectioned. The sections were then dewaxed by xylene, hydrated, and stained with HE Stain Kit (Beyotime, C0105M) or Masson's Trichrome Stain Kit (Solarbio, G1340-7) according to the manual. The sections were then dehydrated through a gradient series of ethanol, cleared by xylene, sealed with neutral resin, and observed under a microscope (Olympus BX53, Tokyo, Japan).

## Starch-iodine tests

Mice were immobilized in jigs and both hind paws were painted with a solution of 3% (wt/vol) iodine in ethanol. Once dry, the paws were painted with a suspension of 40% (wt/vol) starch in mineral oil. Images were taken after 90 s using a microscope (Olympus SZX16, Tokyo, Japan). Sweat was detected as dark spots, and the area of black spots was calculated by Imaris software.

## Analysis of skull and teeth phenotypes

Skull and mandibles were dissected from 6-week-old mice and fixed with 4% paraformaldehyde (PFA) at 4 °C overnight. Samples were stored in PBS (pH 7.4) at 4 °C before being processed. The skull and mandibles were scanned with a Sky Scan1176 (Bruker, Kartuizersweg, Belgium) machine at a spatial resolution of 9 μm for quantitative analysis. Three-dimensional images were reconstructed and analyzed with the scan slice data.

## Statistics and reproducibility

For Fig. 4a, e, f and Supplementary Fig. 1a, pull down assays and gel filtration experiments were repeated three times with similar results. For Fig. 4d, deglycosylation experiments were repeated three times showing same results. For Fig. 5a, EDA cell surface adsorption assay were repeated three times with similar results and representative micrographs are shown. For Fig. 5d, luciferase reporter assays were repeated three times and data are presented as the mean ± SEM with a two-sided Student's *t*-test performed. For Figs. 6a, 6b, 7a, 7c–e and Supplementary Fig. 4, at least three mouse littermates of each indicated genotype were analyzed showing similar results and representative images are shown. For Fig. 6b, c ($n = 7$ adult male mice per group) and 7a, b ($n = 9$ male mice for KO and 10 for other genotypes), representative micrographs are shown and quantification results are presented as mean ± SD with a two-sided Student's *t*-test performed.

## Reporting summary

Further information on research design is available in the Nature Portfolio Reporting Summary linked to this article.

## Data availability

The crystal structure of the EDA·A1$_{THD}$-EDAR$_{CRDS}$ complex was deposited in PDB with accession code 7X9G. All data needed to evaluate the conclusions in the paper are present in the paper and/or the Supplementary Materials. Previously published structures used in this study are: PDB 1RJ7 (Crystal structure of EDA-A1), 1RJ8 (Crystal structure of EDA-A2), 3ALQ (Crystal structure of TNF-TNFR2 complex), 3ME2 (Crystal structure of mouse RANKL-RANK complex) and AF-Q9HAV5 (Alpha Fold structure of EDAR2). Source data are provided with this paper.

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

## Acknowledgements

We thank all the patients and their family members for their participation in this study. We thank the staff members at BL18U1 and BL19U1 of the National Facility for Protein Science in Shanghai (NFPS), China for help with diffraction data collection and processing. We thank Haojie Chen, Jianjie Gu and Chaohua Zheng of the Large-scale Protein Preparation System at the National Facility for Protein Science in Shanghai (NFPS), Zhangjiang Lab for help with mouse breeding. We thank Shufang He, Hong Lu, Shuai Li, Ying Cui, Qian Wang, Rijing Liao of Shanghai Jiao Tong University School of Medicine (SHSMU) for help with confocal microscopy, FACS, mass spectrometry analyses, SPR assays. We thank Xiaowan Lin and Lijun Yan of department of Second Dental Center, Ninth People's Hospital Affiliated with Shanghai Jiao Tong University School of Medicine for help with collection of cohort data. This work was supported by grants from the National Natural Science Foundation of China (31930063 to M.L., 82271004 to Y.W., 31971137 to C.H., 32000843 to Fu.W.), the CAMS Innovation Fund for Medical Sciences (CIFMS) (2019-I2M-5-037 to Y.W.), the National Key Research and Development Program of China (2018YFC2000102 and 2018YFA0107004 to M.L.), the Natural Science Foundation of Shanghai (21ZR1437700 to Y.W.), the Outstanding Academic Leader Program of Science and Technology

Commission of Shanghai Municipality (19XD1422200 to J.W.), the Research Discipline fund from Ninth People's Hospital, Shanghai Jiao Tong University School of Medicine (KQYJXK2020 to Y.W.), and the Project of Biobank of Shanghai Ninth People's Hospital, Shanghai Jiao Tong University School of Medicine (YBKB202101 to Y.W.).

## Author contributions

M.L., Y.W., and J.W. conceived this study. K.Y., C.H., and Fu.W. carried out the bulk of the experiments. K.Y. and Fu.W. crystallized the EDA·A1$_{THD}$-EDAR$_{CRDS}$ complex. Y.K., Fu.W., and J.W. collected diffraction data and determined the crystal structure. J.C., X.L., Y.K., and C.H. constructed the knock-in mice. K.Y., C.J., and Fe.W. analyzed the cohort data. K.Y., C.H., Fu.W., J.C., and C.J. performed animal and molecular experiments. Y.K., C.H., Fu.W., Y.W., and M.L. wrote the manuscript. All authors were involved in data interpretation and writing of the manuscript.

## Competing interests

The authors declare no competing interests.
