## [Peer Review File · Nature Communications]

REVIEWER COMMENTS

Reviewer #1 (Remarks to the Author):

The authors demonstrated that the crystal structure of EDA C-terminal TNF homology domain bound to the N-terminal cysteine rich domains of EDAR. Moreover, mice experiments showed different EDA mutations caused varying degrees of clinical findings. This paper is well written and informative for the readers. However, I have several following comments.

1. In figure 4d, the luciferase reporter assay showed the activity of mutants were decrease than WT. The phenotype of D265G was severe than other mutants and the luciferase activity of D265G was lower than other variants. This results showed that NF- κ B activation of D265G was lower than other variants. However, I wonder if only the level of NF- κ B activation was directly lead to XL-HED phenotype.
2. This comment is similar to comment 1. The authors demonstrated that the clinical findings of D265G mutant were more severe than other mutant mice. In clinical situation, we observed different phenotypes in same mutation. In mice experiments, is there difference of clinical findings in same mutation? If yes, can you think of anything that might have caused it?

Reviewer #2 (Remarks to the Author):

In this paper, the authors present the crystal structure of EDA, a member of the TNF superfamily, in complex with its receptor, the EDA receptor. And the purpose of this paper is to analyze the structural properties of the protein and elucidate its biological activity. The authors have found the crystallization conditions for the complex of EDA protein with the receptor and clarified its structure at a resolution of 2.8 Å. They have also performed SPR analysis of the binding to the receptor and examined the alteration of the interaction with amino acid variants to find the cause of the induction of Hypohidrotic X-linked ectodermal dysplasia, a well-known human genetic disease. In this paper, a mouse model was also created as a proof of concept, and its phenotypes were also analyzed for similarities between the phenotypes that occur in human diseases and those in mice.

Through these diverse experiments, the authors attempt to elucidate the nature of EDA as a cause of human genetic diseases, but it seems as if several studies are needed to strengthen the results.

1. Although the resolution of 2.8 Å is acceptable for structural analysis, I have the impression that the amino acid residues that show important interactions are only partially identified. As a result, data on specific amino acid interactions are poor, and important information on the interaction region is

missing. For example, is there any information on the extent to which water molecules are involved in the interaction? Would it be possible to re-collect data with a higher resolution?

2. Fig. 2 shows the ligand-receptor interaction analysis, especially with mutants, and both A259E and D265G mutants show no interaction with the receptor. On the other hand, the results using mice show that, as phenotypes, the A259E and D265G mutants show little change from wild-type mice for the different phenotypes, especially for D259E. The reason for this can be considered as the difference between mouse and human, but what are the actual changes in the interaction between mouse receptors and ligands? Please provide proof by conducting interaction analysis (SPR or pull-down assay) using mouse-derived ligands and receptors.

3. The discussion in Fig. 6 is a conceptual diagram in which four classifications of EDAs at the site of mutation have been conducted. This concept is acceptable, but we would like to see how the severity of disease associated with each interaction change correlates with its biochemical consequences. This means that if we know the severity of each type and disease, we can determine the impact of the structural changes in the molecule on the disease. For example, how about summarizing the structural changes, the biochemical interaction changes, and the severity of the disease when a Type II mutation occurs, and how to generate mutants in vitro and perform SPR analysis, etc. for each type?

Reviewer #3 (Remarks to the Author):

This is a well written and potentially impactful manuscript that details structural changes in EDA protein, and relates those changes to differing severity of hypohidrotic ectodermal dysplasia (HED) phenotypes in teeth, hair and sweat glands of novel mutant mice, in addition to providing cell biochemical assays that appear to correspond to the mouse phenotype severities. While I am not an expert in structural biology, the categorization of EDA mutation types by location within EDA/EDAR bound structure and correspondence of mouse phenotype severity with biochemical assays appears logical and potentially will be of help for genetic counseling and prognosis of patients diagnosed with HED. Overall this work contributes significantly to the body of knowledge on HED and causes of HED.

Comments:

An NF-kb luciferase assay in a cell line that does not express endogenous ERAD could be artifactual. Therefore this data should ideally be supported by additional evidence that the mutant EDA proteins are defective to different degrees in driving NF/kb activity.

Only male mice were studied. Some discussion regarding expected findings in females is warranted.

Taurodontism severity can be quantified using previously established quantifiable measures of the taurodontism phenotype. Use of such a quantifiable method would strengthen the data and also confirm the reported incidence provided.

There is no mention of presence/absence skull or facial dysmorphologies, though these have been reported in humans with ED and HED. A short statement or summary regarding craniofacial phenotype of the mice plus discussion of why these are or are not present in the mutant mice as compared to humans is desirable.

Point-to-point responses to reviewers' comments

Structural insights into pathogenic mechanism of hypohidrotic ectodermal dysplasia caused by *EDA* variants

NCOMMS-22-25815

Reviewer #1:

The authors demonstrated that the crystal structure of EDA C-terminal TNF homology domain bound to the N-terminal cysteine rich domains of EDAR. Moreover, mice experiments showed different EDA mutations caused varying degrees of clinical findings. This paper is well written and informative for the readers. However, I have several following comments.

Thanks!

Minor points:

1. In the manuscript figure 4d, the luciferase reporter assay showed the activity of mutants were decrease than WT. The phenotype of D265G was severe than other mutants and the luciferase activity of D265G was lower than other variants. This results showed that NF- κ B activation of D265G was lower than other variants. However, I wonder if only the level of NF- κ B activation was directly lead to XL-HED phenotype.

We thank the reviewer for this good point. The EDA signaling pathway consists of EDA, its receptor EDAR, and an adaptor protein EDARADD and its activation leads to NF- κ B mediated transcription [1, 2]. The comparison of NF- κ B-LacZ reporter activation in *tabby* versus wild type mice showed that decreased EDA signaling effectively induces a severe decrease in NF- κ B activation [3]. The relevance of NF- κ B activation was further confirmed when mice with suppressed NF- κ B activity were shown to exhibit HED phenotypes, strikingly similar to EDA (*tabby*) and EDAR (*downless*) mutant mice [4-6]. Notably, mutations in components of the NF- κ B pathway such as TRAF6 and IKK have been identified in HED patients [7, 8]. Moreover, supply of recombinant EDA restores placodes

and NF- κ B signals in *Eda*-deficient skin, but not in *I κ B α Δ N* (non-degradable I κ B α , inhibition of NF- κ B pathway) transgenic skin [3], indicating that NF- κ B activation is essential for the EDA signaling [9]. Taken together, these results suggest that NF- κ B pathway is a major transducer of the EDA signaling[10].

2. This comment is similar to comment 1. The authors demonstrated that the clinical findings of D265G mutant were more severe than other mutant mice. In clinical situation, we observed different phenotypes in same mutation. In mice experiments, is there difference of clinical findings in same mutation? If yes, can you think of anything that might have caused it?

In our study, mice with same *Eda* mutations always exhibited similar phenotypes. We speculate that in clinical situations, different phenotypes may be documented for the same mutation due to the following reasons: 1) HED is mainly caused by mutational inactivation of the EDA, EDAR or EDAR, which encodes the ectodysplasin pathway responsible for formation of tissues of ectodermal origin [11]. Other general morphogenetic signaling pathways such as Wnt, BMP and Shh also play important roles in regulating the formation of skin appendages including teeth, hair follicles and exocrine glands [12], thus increasing the complexity of the pathogenic mechanisms underlying HED; 2) The genetic backgrounds in XL-HED patients are often significantly different from each other, and many affected individuals may harbor other mutations that aggravate the HED phenotypes. In contrast, laboratory mice show very subtle genetic variations; 3) The abnormal sweating ability and sparse hair phenotype of patients are often subjective judgments of clinicians, and their judgment criteria may not be the same.

Reviewer #2:

In this paper, the authors present the crystal structure of EDA, a member of the TNF superfamily, in complex with its receptor, the EDA receptor. And the purpose of this paper is to analyze the structural properties of the protein and elucidate its biological activity. The authors have found the crystallization conditions for the complex of EDA protein with the receptor and clarified its structure at a resolution of 2.8 Å. They have also performed SPR

analysis of the binding to the receptor and examined the alteration of the interaction with amino acid variants to find the cause of the induction of Hypohidrotic X-linked ectodermal dysplasia, a well-known human genetic disease. In this paper, a mouse model was also created as a proof of concept, and its phenotypes were also analyzed for similarities between the phenotypes that occur in human diseases and those in mice.

Through these diverse experiments, the authors attempt to elucidate the nature of EDA as a cause of human genetic diseases, but it seems as if several studies are needed to strengthen the results.

Thanks!

1. Although the resolution of 2.8 Å is acceptable for structural analysis, I have the impression that the amino acid residues that show important interactions are only partially identified. As a result, data on specific amino acid interactions are poor, and important information on the interaction region is missing. For example, is there any information on the extent to which water molecules are involved in the interaction? Would it be possible to re-collect data with a higher resolution?

We thank the reviewer for pointing out this issue. We have carefully reexamined the electron density map of the EDA·A1_{THD}-EDAR_{CRDS} complex, and we believe that the quality of the current dataset is good. The interaction between EDA·A1_{THD} and EDAR_{CRDS} is quite stable, and the electron density map clearly reveals the important amino acids at the interface (Reviewer Figure 1a-d). There are no distinct direct or different peaks in the electron density map that could be traced as water molecules involved in the EDA·A1-EDAR interaction. Although water molecules are found outside the interaction region, they are likely not essential to the ligand-receptor interaction. We have noticed that some amino acids in EDAR_{CRDS} were unresolved in our structure model (residues 60–65 and 119-126), probably due to the high flexibility of the corresponding loops. However, these residues are not at the EDA-EDAR interface (Reviewer Figure 1f), suggesting that these missing amino acids unlikely make major contribution to the ligand-receptor interaction. Taken together, we conclude that our crystal structure reveals the molecular basis for the interaction between EDA·A1_{THD} and EDAR_{CRDS}.

Following this reviewer's suggestion, we have prepared new batches of purified protein complex and made every effort to improve the crystal structure quality, including optimizing crystallization and cryo-protectant conditions, testing crystal dehydration, as well as trying different annealing time. Unfortunately, these efforts did not improve the diffraction limit and no dataset with higher resolution was obtained.

Reviewer Figure 1: Representative experimental density map. **a-b** The Sigma-A weighted 2Fo-Fc electron density maps show the interface of EDA and EDAR near to the residues Ala259 (a) and Asp265 (b) and no more peaks in the raw data could be traced in this area. Contours are presented at the 1.0 σ level. **c** The refined structure is drawn as b-factor putty and the EDA-EDAR interface is denoted in the red circle. **d** The structure of EDAR_{CRDS} adopts an elongated conformation. Some amino acids in EDAR_{CRDS} (residues 60–65 and 119–126) were unresolved in our structure due to their high flexibility. These residues are not at the interface region and they are oriented in a direction away from the EDA molecule. preEDAR: Superposition of the predicted EDAR_{CRDS} generated by AlphaFold (PDB: AF-Q9UNE0) onto

the EDA·A1_{THD}-EDAR_{CRDS} complex.

2. Fig. 2 shows the ligand-receptor interaction analysis, especially with mutants, and both A259E and D265G mutants show no interaction with the receptor. On the other hand, the results using mice show that, as phenotypes, the A259E and D265G mutants show little change from wild-type mice for the different phenotypes, especially for D259E. The reason for this can be considered as the difference between mouse and human, but what are the actual changes in the interaction between mouse receptors and ligands? Please provide proof by conducting interaction analysis (SPR or pull-down assay) using mouse-derived ligands and receptors.

Following this reviewer's suggestion, we have investigated the interaction between mouse EDA and EDAR using a pull-down assay and obtained similar results to those with human ligands and receptors. As shown in Reviewer Figure 2, both the A259E and D265G mutations in mouse EDA protein severely disrupt the interaction with mouse EDAR, whereas the R276C mutation slightly weakens the interaction with the receptor. We have included this data in the revised Supplementary Fig. 3e and modified the text as follows:

“The effect of mutations interrupting the binding of EDAR was the same for both forms of EDA·A1_{THD} (Fig. 4a and Supplementary Fig. 3d), and similar results were obtained with mouse EDA·A1_{THD} and EDAR_{CRDS} (Supplementary Fig. 3e).” (Page 10 Line 220 – Line 223)

EDA provides an inductive rather than a maintenance signal during ectodermal development [13, 14]. Although biochemical experiments showed that both A259E and D265G mutations disrupt the interaction between EDA and EDAR, we cannot rule out that the mutant EDA still remains very weak interaction with the receptor in vivo to activate the NF- κ B signaling pathway to a certain degree, which is not detectable by the pull-down assay. In fact, as revealed by the luciferase reporter assay, both A259E and D265G mutations substantially impair but not completely abolish the transcriptional activation of the NF- κ B pathway (revised Fig. 4D), indicating that the reporter assay is a more sensitive technique for evaluating the cellular defect of EDA·A1-EDAR interacting mutations. Notably, the fact that the transcriptional activation defect of the D265G mutant is more

prominent than that of A259E is consistent with the severity of HED phenotypes in mutant mice.

In addition, the observation that A259E and D265G mutants show little change from wild-type mice implies that mouse ectodermal development depends on the EDA signaling less than that of humans. Consistently, patients harboring *EDA null* mutation exhibit severe HED characters (anodontia, scalp hair and eyebrow absent or sparse), whereas *Eda* knockout mice exhibit milder phenotypes than humans (only the third molar is missing, and scalp hair was slightly affected).

Reviewer Figure 2 (revised Supplementary Fig. 3e): Pull-down assay using anti-Flag beads with ectopically expressed mouse Flag-EDA-A1_{THD} and purified EDAR_{CRDS}-MBP. The levels of each protein in the input and pull-down samples were analyzed by immunoblotting with the indicated antibodies. The purified EDA-A1_{THD} ectodomain showed two close bands in western blots, with the higher molecular-weight band corresponding to a glycosylated form of EDA-A1_{THD} as revealed by a deglycosylation analysis with peptide N-glycanase (PNGase) F (revised Supplementary Fig. 3c).

3. The discussion in Fig. 6 is a conceptual diagram in which four classifications of EDAs at the site of mutation have been conducted. This concept is acceptable, but we would like to see how the severity of disease associated with each interaction change correlates with its biochemical consequences. This means that if we know the severity of each type and disease, we can determine the impact of the structural changes in the molecule on the disease. For example, how about summarizing the structural changes, the biochemical interaction changes, and the severity of the disease when a Type II mutation occurs, and how to generate mutants in vitro and perform SPR analysis, etc. for each type?

We thank the reviewer for this constructive comment. Following this reviewer's suggestion, we have carefully analyzed the structural changes corresponding to the pathogenic missense mutations in EDA·A1_{THD}. Based on the idea that the location of a mutation as well as the change in physico-chemical properties of the residue contribute to its biochemical consequences, we have summarized the severity of the deleterious effects caused by *EDA* mutations in the revised Supplementary Table 2, and found that the severity of diseases (syndromic HED vs non-syndromic NSTA) largely correlate with the predicated mutation effect.

We have selected some mutations from Type I (L271P, G291R, Y320C), Type II (Y304C, R334H, F379S) and Type IV (G299R, D316G, T338M), generated the corresponding mutants in vitro, and examined the impact of these mutations on protein expression, the EDA-EDAR interaction and the activation of the NF-κB signaling (Reviewer Figure 3a-f). Among Type I mutations, a proline substitution in position 271 (L271P) alters the local geometry of the A'A'' loop (Reviewer Figure 3a). Gly291 and Tyr320 are buried inside the jelly roll β-sandwich, and mutations of G291R and Y320C reduce the hydrophobic interaction which may impair the protein folding (Reviewer Figure 3a). Moreover, the G291R mutation introduces a bulky sidechain that could lead to a clash with the peripheral sidechains (Reviewer Figure 3a). Collectively, these three mutations likely lead to the instability of the EDA protein and the disruption of the NF-κB signaling, which were validated by the western blot analysis and the luciferase reporter assay (Reviewer Figure 3d and e). Consistently, those affected patients containing L271P, G291R or Y320C exhibit congenital syndromic HED phenotypes (revised Supplementary Table 2). In Type II group, Tyr304 and Phe379 are close in space, lying in a hydrophobic environment formed by Leu330, Ala349 and Val351 in an adjacent monomer (Reviewer Figure 3b). Mutations of Y304C and F379S would disrupt these packing interactions among the monomers, leading to the destabilization of the EDA trimer. As to the R334H mutation that locates at the terminal end of the Eβ strand, the histidine substitution has little impact on charge and steric hindrance (Reviewer Figure 3b). Consistently, western blot analysis and NF-κB luciferase reporter assay showed that R334H causes mild defects compared to the Y304C and F379S mutations (Reviewer Figure 3d and e), and patients harboring R334H mutation exhibit non-syndromic NSTA (revised Supplementary Table 2).

In Type IV, mutations of G299R, D316G and T338M are dispersed at the EDA exterior region and the consequences of structural changes are difficult to evaluate (Reviewer Figure 3c). However, results of Western blot analysis and NF- κ B luciferase reporter assay on these mutations are consistent, in which G299R induces protein destabilization and causes more severe defects in the activation of the NF- κ B pathway than the D316G and T338M mutations (Reviewer Figure 3d and e). In addition, affected individuals carrying the G299R mutation display syndromic HED phenotypes, whereas patients with D316G or T338M show mild non-syndromic manifestations. Moreover, pull-down assay showed that both D316G or T338M mutations, as well as Type II mutation R334H only slightly impair the EDA-EDAR interaction (Reviewer Figure 3f). This is in align with the result of the luciferase reporter assay on these mutations and with the mild manifestations in the corresponding patients (Reviewer Figure 3e, revised Supplementary Table 2). Taken together, the findings of our study provide structural insights into the pathogenic mechanism of a large part of disease-causing *EDA* variants.

Reviewer Figure 3: Structural changes and biochemical consequences of the pathogenic missense mutations in EDA·A1_{THD}. **a-c:** Structural change in nine EDA·A1 mutants with the mutated residues showing in stick model and colored in red. **d** Protein expression of Flag-EDA·A1_{THD} variants detected by Western blot. HEK293T cells were seeded in a 12-well plate

the day before transfection. Transfections were performed using 1 µg of vectors (pcDNA3.1) encoding WT or mutant Flag-tagged EDA-A1_{THD} and Lipofectamine 3000 (Invitrogen). 48 hours after the transfection, cell extracts were harvested and subjected to Western blot analysis with the indicated antibodies. **e** Luciferase reporter assay showed that the mutant EDA proteins were defective in driving NF-κB activity. HEK293T cells stably expressing full-length EDAR were seeded in 12-well dishes one day before transfection. 500 ng of pNF-κB Luc plasmid (Promega, E8491) and 10 ng of pRL-TK Renilla reference plasmid (Promega, E2231) were co-transfected for each well using Lipofectamine 3000 (Invitrogen). 24 hours after the transfection, an equivalent volume of the supernatant of WT or mutated EDA-A1_{THD} proteins were added to each well, respectively. After 12 hours, Firefly luciferase activity in the cell lysates was measured and normalized to Renilla luciferase activity using a dual-luciferase reporter assay system. ***P < 0.001. **f** Pull-down assay using anti-Flag beads with ectopically expressed human Flag-EDA-A1_{THD} and purified EDAR_{CRDS}-MBP. Those mutations (D316G, R334H and T338M) that have no significant effect on protein expression in (b) are selected. The levels of each protein in the input and pull-down samples were analyzed by immunoblotting with the indicated antibodies. WT, wild type; MBP (EDAR), anti-MBP to detect EDAR_{CRDS}-MBP; Flag (EDA), anti-Flag to detect Flag-EDA-A1_{THD}.

Reviewer #3:

This is a well written and potentially impactful manuscript that details structural changes in EDA protein, and relates those changes to differing severity of hypohidrotic ectodermal dysplasia (HED) phenotypes in teeth, hair and sweat glands of novel mutant mice, in addition to providing cell biochemical assays that appear to correspond to the mouse phenotype severities. While I am not an expert in structural biology, the categorization of EDA mutation types by location within EDA/EDAR bound structure and correspondence of mouse phenotype severity with biochemical assays appears logical and potentially will be of help for genetic counseling and prognosis of patients diagnosed with HED. Overall this work contributes significantly to the body of knowledge on HED and causes of HED.

Thanks!

1. An NF-κB luciferase assay in a cell line that does not express endogenous EDAR could be artifactual. Therefore, this data should ideally be supported by additional evidence that

the mutant EDA proteins are defective to different degrees in driving NF/κB activity.

We thank the reviewer for pointing out this issue. Endogenous EDAR is expressed in HaCaT cells as shown in the database website (<https://www.proteinatlas.org/>), and as verified by the RT-PCR analysis (Reviewer Figure 4a). We have performed the luciferase reporter assay with HaCaT cells and observed that mutant EDA displays a defect in driving NF-κB transcriptional activation compared to WT (Reviewer Figure 4b). Among the three mutations (D265G, A259E and R276C), the D265G mutation severely impairs the activation of downstream signaling pathways, whereas mutations of A259E and R276C only cause mild defects. This observation is consistent with the result using HEK293T cells that ectopically express full-length EDAR. We have included this data in the revised Supplementary Fig. 3g and modified the text as follows.

“The mutation effect of EDA-A1 in driving NF-κB transcriptional activation was further confirmed using HaCaT cells that express endogenous EDAR (Supplementary Fig. 3g).”
(Page 11 Line 245 – Line 247)

Reviewer Figure 4: The interaction between EDAR_{CRDS} and EDA-A1_{THD} variants was assessed by a luciferase reporter assay with HaCaT cells. **a** RT-PCR analysis showing the expression of endogenous EDAR in HaCaT cells. HEK293T cDNA sample was used as a control. **b** Luciferase reporter assay using HaCaT cells showed that the mutant EDA proteins were defective to different degrees in driving NF-κB activity. pNF-κB Luc plasmid and pRL-TK Renilla reference plasmid were co-transfected into HaCaT cells. After 24 hours, an equivalent amount of WT or mutant EDA-A1_{THD} proteins were added. After 18 hours, Firefly luciferase activity in the cell lysates was measured and normalized to Renilla luciferase activity.

2. Only male mice were studied. Some discussion regarding expected findings in females is warranted.

Following this reviewer's suggestion, female mice with homozygous *Eda* mutations were analyzed and similar HED phenotypes to that in male mice were observed (Reviewer Figure 5). We have included the data in the revised Supplementary Fig. 6 and modified the text as follows.

“The *EDA* gene is located on the X chromosome and male carriers of *EDA* variants (hemizygous for an *EDA* mutation) tend to be more affected than heterozygous female carriers who may have mild-to-asymptomatic manifestations due to somatic mosaicism [15]. We also analyzed homozygous *Eda* mutant female mice and observed similar HED phenotypes to those in hemizygous male mice (Supplementary Fig. 6), suggesting that *EDA* mutations affect ectodermal development in both male and female mice.” (Page15 Line 332 – Line 338)

Reviewer Figure 5 (revised Supplementary Fig. 6): Deficiency of EDA causes similar ectodermal dysplasia in female mice to that of hemizygous male mice. The *Eda* homozygous mutant female mice were analyzed for their ectodermal derivatives. The *Eda*^{KO/KO} mice showed the most severe defects characterized by hairless tails and abdomen, kinked tail tips, a bald patch behind ears and abnormal eyelid development. The *Eda*^{KO/KO} mice had only two mandibular molars, with lower M1 having only two rounded cusps and M2 exhibiting taurodontism. The *Eda*^{KO/KO} mice showed no eccrine sweat glands in the footpads and displayed defects in sweating function revealed by the Starch-iodine test. The HED phenotypes of the *Eda*^{D265G/D265G} mice were milder than that in the *Eda*^{KO/KO} mice, showing tooth agenesis, defective sweating abilities, scanty abdomen hair and ear hair, but with normal tail hair and tips. Although three mandibular molars were present in the *Eda*^{D265G/D265G} mutant mice, lower M1 was small with only four of normal seven cusps formed, and M2 also exhibited

taurodontism. The *Eda*^{A259E/A259E} and *Eda*^{R276C/R276C} mutant mice exhibited very mild tooth agenesis with molars displaying flattened, rounded cusps but no obvious abnormalities in tail tips, eyelid, tail hair, abdomen hair and hair behind ears.

3. Taurodontism severity can be quantified using previously established quantifiable measures of the taurodontism phenotype. Use of such a quantifiable method would strengthen the data and also confirm the reported incidence provided.

We thank the reviewer for this good point. We have quantified the taurodontism phenotypes in *Eda* mutant mice as suggested (Reviewer Figure 6), and have included this data in the revised Supplementary Fig. 5b. According to the diagnostic criteria of human taurodontism proposed by Shifman et al [16], the ratio of the distance AB (between the lowest point of the pulp roof A and the highest point of the pulp floor B) to the distance AC (between the lowest point of the pulp roof A and the apex of the longest tooth root C), and the distance BD (from the highest point of the pulp floor B to the enamel-cemental junction D) are calculated to evaluate the severity of taurodontism.

Reviewer Figure 6 (revised Supplementary Fig. 5b): Quantification of the taurodontism phenotype in *Eda* mutant mice. Top: Schematic diagram of the taurodontism phenotype. Taurodontism is characterized by an elongation of the pulp chamber extending into the root area. A, pulp roof; B, pulp floor; C, apex of the longest tooth root; D, enamel-cemental junction.

Bottom: Quantification of taurodontism phenotype. n=7 adult (about 6-week old) male mice per group; ***P <0.001.

4. There is no mention of presence/absence skull or facial dysmorphologies, though these have been reported in humans with ED and HED. A short statement or summary regarding craniofacial phenotype of the mice plus discussion of why these are or are not present in the mutant mice as compared to humans is desirable.

Thanks for this point. We have analyzed the craniofacial phenotypes in the *Eda* knock-out mice and found no obvious craniomaxillofacial abnormality (Reviewer Figure 7). We have included this data in the revised Supplementary Fig. 5c and revised the text as follows.

“Although skull and facial dysmorphologies have been reported in human patients suffering from HED [17, 18], we observed no substantial craniomaxillofacial deformities in *Eda* knock-out mice (Supplementary Fig. 5c). This result indicates that craniomaxillofacial development in different species depends on the EDA function in varying degrees and mouse craniomaxillofacial development exhibits less vulnerable to the deficiency in EDA signaling.” (Page 14 Line 327 – Page 15 Line 332)

Reviewer Figure 7 (revised Supplementary Fig. 5c): Micro-computed tomography (μCT) analysis of craniofacial phenotypes in *Eda* knockout and WT mice. No obvious craniomaxillofacial abnormality was observed in *Eda* knockout mice.

References

1. Reyes-Realí, J., et al., *Hypohidrotic ectodermal dysplasia: clinical and molecular review*. International journal of dermatology, 2018. **57**(8): p. 965-972.
2. Mikkola, M.L., *TNF superfamily in skin appendage development*. Cytokine Growth Factor Rev, 2008. **19**(3-4): p. 219-30.
3. Schmidt-Ullrich, R., et al., *NF-kappaB transmits Eda A1/EdaR signalling to activate Shh and cyclin D1 expression, and controls post-initiation hair placode down growth*. Development (Cambridge, England), 2006. **133**(6): p. 1045-57.
4. Cascallana, J., et al., *Ectoderm-targeted overexpression of the glucocorticoid receptor induces hypohidrotic ectodermal dysplasia*. Endocrinology, 2005. **146**(6): p. 2629-38.
5. Gugasyan, R., et al., *The transcription factors c-rel and RelA control epidermal development and homeostasis in embryonic and adult skin via distinct mechanisms*. Molecular and cellular biology, 2004. **24**(13): p. 5733-45.
6. Schmidt-Ullrich, R., et al., *Requirement of NF-kappaB/Rel for the development of hair follicles and other epidermal appendices*. Development (Cambridge, England), 2001. **128**(19): p. 3843-53.
7. Wisniewski, S. and W. Trzeciak, *A rare heterozygous TRAF6 variant is associated with hypohidrotic ectodermal dysplasia*. The British journal of dermatology, 2012. **166**(6): p. 1353-6.
8. Johnston, A., et al., *A Novel Mutation in IKBKG/NEMO Leads to Ectodermal Dysplasia with Severe Immunodeficiency (EDA-ID)*. Journal of clinical immunology, 2016. **36**(6): p. 541-3.
9. Mikkola, M.L. and I. Thesleff, *Ectodysplasin signaling in development*. Cytokine & Growth Factor Reviews, 2003. **14**(3-4): p. 211-224.
10. Sadier, A., et al., *The ectodysplasin pathway: from diseases to adaptations*. Trends Genet, 2014. **30**(1): p. 24-31.
11. Cluzeau, C., et al., *Only four genes (EDA1, EDAR, EDARADD, and WNT10A) account for 90% of hypohidrotic/anhidrotic ectodermal dysplasia cases*. Human Mutation, 2011. **32**(1): p. 70-72.
12. Mikkola, M.L., *Genetic basis of skin appendage development*. Seminars In Cell & Developmental Biology, 2007. **18**(2): p. 225-236.
13. Gaide, O. and P. Schneider, *Permanent correction of an inherited ectodermal dysplasia with recombinant EDA*. Nature medicine, 2003. **9**(5): p. 614-8.
14. Kowalczyk, C., et al., *Molecular and therapeutic characterization of anti-ectodysplasin A receptor (EDAR) agonist monoclonal antibodies*. The Journal of biological chemistry, 2011. **286**(35): p. 30769-30779.

15. Anbouba, G.M., E.P. Carmany, and J.L. Natoli, *The characterization of hypodontia, hypohidrosis, and hypotrichosis associated with X-linked hypohidrotic ectodermal dysplasia: A systematic review*. Am J Med Genet A, 2020. **182**(4): p. 831-841.
16. Shifman, A. and I. Chanannel, *Prevalence of taurodontism found in radiographic dental examination of 1,200 young adult Israeli patients*. Community Dentistry and Oral Epidemiology, 1978. **6**(4): p. 200-203.
17. Goodwin, A., et al., *Craniofacial morphometric analysis of individuals with X-linked hypohidrotic ectodermal dysplasia*. Molecular genetics & genomic medicine, 2014. **2**(5): p. 422-9.
18. Sonnesen, L., et al., *Upper cervical spine and craniofacial morphology in hypohidrotic ectodermal dysplasia*. European archives of paediatric dentistry : official journal of the European Academy of Paediatric Dentistry, 2018. **19**(5): p. 331-336.

REVIEWERS' COMMENTS

Reviewer #1 (Remarks to the Author):

I have no further comment.

Reviewer #2 (Remarks to the Author):

I am pleased with the authors' courteous response. I believe that the content has been improved to the point where it is worthy of publication.

Reviewer #3 (Remarks to the Author):

The authors responded strongly to my comments and suggestions, including generation of additional data to support their conclusions. Assuming the other reviewers (particularly those who have expertise in structural biology) agree, the manuscript is now appropriate for publication.

Point-to-point responses to reviewers' comments

Structural insights into pathogenic mechanism of hypohidrotic ectodermal dysplasia caused by *Ectodysplasin A* variants

NCOMMS-22-25815

Reviewer #1 (Remarks to the Author):

I have no further comment.

Thanks!

Reviewer #2 (Remarks to the Author):

I am pleased with the authors' courteous response. I believe that the content has been improved to the point where it is worthy of publication.

Thanks!

Reviewer #3 (Remarks to the Author):

The authors responded strongly to my comments and suggestions, including generation of additional data to support their conclusions. Assuming the other reviewers (particularly those who have expertise in structural biology) agree, the manuscript is now appropriate for publication.

Thanks!